# Identification of *Hydatigera* Species in Wildcats (*Felis silvestris*) from Central Spain

**DOI:** 10.3390/ani15223340

**Published:** 2025-11-19

**Authors:** Pablo Matas-Méndez, Lorena Esteban-Sánchez, Francisco Ponce-Gordo, Marta Mateo-Barrientos

**Affiliations:** 1Facultad de Veterinaria, Universidad Alfonso X El Sabio, Villanueva de la Cañada, 28691 Madrid, Spain; 2Departamento de Microbiología y Parasitología, Facultad de Farmacia, Universidad Complutense de Madrid, 28040 Madrid, Spain; lorees01@ucm.es (L.E.-S.); mmateo14@ucm.es (M.M.-B.)

**Keywords:** *Felis silvestris*, *Hydatigera taeniaeformis* complex, *Hydatigera kamiyai*, European *Hydatigera* sp., PCR multiplex

## Abstract

The European wildcat (*Felis silvestris*) is a wild feline found in various regions of Spain, with a diet primarily consisting of mice and rabbits. These animals can become infected with intestinal parasites, such as tapeworms of the genus *Hydatigera*. Some of these parasites are very difficult to distinguish under a microscope but can be identified using DNA-based methods. In this study, 26 road-killed wildcats from central Spain were examined, and 73% were found to be infected with *Hydatigera* tapeworms. Using molecular tools, researchers identified two different *Hydatigera* species never before reported in Spanish wildcats. A rapid genetic test was also developed to differentiate them. This study extends the known geographical range of the species in the *H. taeniaeformis* complex (*Hydatigera kamiyai* and an unnamed *Hydatigera* sp.) in Europe and provides a reliable molecular tool for identifying them, which is essential for further epidemiological studies.

## 1. Introduction

The European wildcat (*Felis silvestris* Schreber, 1777) is a mesocarnivore with a broad distribution across Europe, whose populations have been increasing in specific regions of Central Europe [1]. In Spain, two subspecies are recognized: the European subspecies (*F. silvestris silvestris* Miller, 1912), found in the northern part of the country, north of the Duero and Ebro rivers, and the Iberian subspecies (*F. silvestris tartessia* Miller, 1912), located in the southern part of the peninsula, specifically in Doñana National Park [2,3,4]. Populations on the Iberian Peninsula are in decline [5], due to multiple factors including habitat fragmentation, competition and hybridization with domestic cats (*Felis catus* Linnaeus, 1758), and anthropogenic pressures such as poisoning and roadkills [6]. The wildcat’s diet primarily consists of rodents (mice and voles) and lagomorphs (rabbits and hares) [7].

Among the most common parasites infecting these animals are nematodes and cestodes. Within the cestodes, the most frequently reported belong to the genus *Hydatigera* Lamarck, 1816 [8,9]. This genus was previously considered a synonym of *Taenia* by Verster in 1969 [10], but was reinstated by Nakao and coworkers in 2013 [11] to include the species *Hydatigera krepkogorski* (Schulz and Landa, 1934), *Hydatigera parva* (Baer, 1924), and *Hydatigera taeniaeformis* (Batsch, 1786). All of these species are found in felids; *H. krepkogorski* also occurs in canids, *H. parva* in herpestids, mustelids, and viverrids; and *H. taeniaeformis* also in viverrids and canids [10,12,13,14,15]. The latter species has long been regarded as a species complex based on the analysis of mitochondrial sequences, with isolates collected in Turkey, Finland and Japan showing genetic differences from others collected in Belgium, Australia, Kazahkstan, Malaysia, China, and also Japan [11,16,17]. The complete mitochondrial genome of *T. taeniaeformis* of Chinese origin [18] and of German origin [19] showed significant differences, and Galimberti et al. [20] identified three different putative species within *H. taeniaeformis*, which were ultimately named by Lavikainen et al. [21] as *H. taeniaeformis* sensu stricto (s.s.), *Hydatigera kamiyai* Lavikainen et al., 2016, and an undescribed species from European felids currently referred to as *Hydatigera* sp. Recently, two new putative species have been identified in China from rodent hosts [22,23]. However, these have only been described from larval stages, and the adult cestodes of the Asian *Hydatigera* spp. have not yet been described; therefore, their inclusion within the *Hydatigera taeniaeformis* complex remains uncertain.

The host range of *H. taeniaeformis* s.s., *H. kamiyai* and the European *Hydatigera* sp. based on genetic data include felids as hosts for all of them, while rodents (murids) are intermediate hosts of *H. taeniaeformis*, rodents (murids and other rodent groups) are intermediate hosts of *H. kamiyai*, and the intermediate host(s) are yet not determined for the European *Hydatigera* sp. (Table 1). Morphological differentiation of the adult cestodes among the three species described by Lavikainen et al. within the *H. taeniaeformis* sensu lato (s.l.) complex is extremely difficult [21], but molecular analyses enable their discrimination [16,20,21,24,25].

There are almost no previous data on cestode infections in wildcats in Spain; the only data available (referred to *H. taeniaeformis* s.l.) are those obtained more than 30 years ago by Torres et al. [26] in samples from all over Spain, who found a overall 60.3% prevalence; and those obtained between 2008–2015 by Gómez-Galindo et al. [27] in samples collected in the south-east of Spain, who found a 36.8% prevalence. Other cestodes found in these studies included *Taenia pisiformis*, *Joyeuxiella pasqualei*, *Diplopylidium nolleri* and *Mesocestoides* spp. In both studies, identifications were based on morphological characters. To date, no studies have described the clinical manifestations of *Hydatigera* infection in European wildcats, and data in domestic cats are also scarce; in general, as with most intestinal cestodes in felids, infections by adult *Hydatigera* tapeworms are considered largely subclinical [8,9]. However, a few isolated clinical cases have been reported in domestic cats, including an acute intestinal obstruction caused by (*Taenia*) *Hydatigera taeniaeformis* s.l., which required surgical removal of the tapeworms [28]. This highlights that, although rare, heavy infections may lead to clinical disease. From an epidemiological perspective, identifying *Hydatigera* species is important because they differ in their life cycles, intermediate hosts, and geographic distributions, thus providing insights into trophic relationships and potential transmission pathways between wild and domestic carnivores. There is an important gap in the literature on *Hydatigera* infections in Spanish wildcats, and the objective of this study is provide new, recent data on the presence and distribution of this cestode genus in the Spanish European wildcats.

**Table 1 animals-15-03340-t001:** Available data on hosts and geographic distribution of *Hydatigera taeniaeformis* sensu stricto, *Hydatigera kamiyai*, and European *Hydatigera* sp.

Parasite Species	Definitive Host	Intermediate Host	Country	References
*H. taeniaeformis* s.s.		Muridae	Japan	[16]
	Felidae (*Felis silvestris catus*)	Not indicated	Australia	[29]
		Muridae	India	[30,31]
	Felidae (*Felis silvestris catus*)		Korea	[32]
	Felidae (*Prionailurus bengalensis*)	Not indicated	China	[24]
	Canidae		Switzerland	[14]
	Felidae		Australia	[33]
		Muridae	India	
	Canidae (*Canis lupus familiaris*)		Germany	[34]
	Felidae (*Felis silvestris catus*)			
		Muridae	Kazakhstan, Turkey	[17]
	Canidae (*Canis lupus familiaris*)		Japan	[15]
		Not indicated	Belgium	[11]
	Felidae (*Felis silvestris catus*) Stool		USA	[35]
		Muridae	Serbia	[36]
	Felidae (*Felis silvestris catus*)		Mexico	[37]
	Felidae *(Leopardus geoffroyi)*		Brazil	[38]
	Not indicated		Finland	[11]
			Japan	
		Muridae	Senegal	[39]
		Muridae	Spain	[21]
*H. kamiyai*	Felidae (*Felis silvestris catus*)		Finland, France, Australia	[21]
	Felidae (*Felis silvestris silvestris*)		Italy	
	Felidae (*Prionailurus bengalensis*)		Russia	
		Muridae	Bosnia, Latvia, Russia, Cambodia, Laos, Thailand, Vietnam, Ethiopia, South Africa	
		Cricetidae	Finland, Norway, Russian, Sweden	
		Cricetidae	Poland	[40]
		Cricetidae, Muridae, Soricidae	Luxembourg	[41]
	Felidae (*Felis silvestris silvestris*)			[42]
		Cricetidae, Muridae	Serbia	[36]
	Felidae (*Felis silvestris silvestris*)		Germany	[43]
	Felidae (*Panthera leo*)		Namibia	[44]
		Cricetidae	China	[45]
		Cricetidae, Muridae	Czech Republic	[46]
	Not indicated		France	[47]
		Nesomyidae	United Kingdom	[48]
*Hydatigera* sp.	Felidae (*Felis silvestris catus*)		France	[21]
	Felidae (*Felis silvestris silvestris*)		Italy	[20]

## 2. Materials and Methods

### 2.1. Sample Origin

Over a period of 36 months (January 2021–December 2023), a total of 26 road-killed European wildcats (*F. silvestris*) were collected from seven provinces across two autonomous communities (Castilla-La Mancha and Castilla y León) in central Spain. The carcasses were transported by environmental officers to regional wildlife recovery centers, where they were stored frozen at −20 °C to ensure proper preservation. The collection of these and other road-killed carnivores was conducted under authorization from the regional environmental departments (permits: DGPFEN/SEN/avp_21_103_bis for Castilla-La Mancha and AB/is. Exp.AUES/CYL/001/2021 for Castilla y León). Age estimation of the individuals was based on body size, weight, and dentition, and all specimens were classified as adults (>1 year). The classification of specimens as adults (over one year old) was based primarily on dental analysis, supplemented by body size and weight. Dental examination confirmed complete eruption of the permanent dentition. It is essential to note that the apical foramen of the canine root was closed and that the teeth showed incipient wear on the cusps of the canines and incisors. This level of dental wear, combined with evidence of a fully mature and closed tooth root, constitutes a non-invasive criterion established in the literature on wildcats for reliably classifying an individual in the >1 year age class [49,50].

### 2.2. Initial Processing and Cestode Recovery

Necropsies were performed on the 26 individuals. The intestinal package was extracted, and its contents were washed several times with distilled water using sieves. Cestodes retrieved from the small intestine were washed and preserved in 70% ethanol at 4 °C until morphological identification and DNA extraction.

As many strobilae were fragmented, the number of cestodes per individual was estimated by counting scoleces. All cestodes with identifiable scolex and strobila belonging to the family Taeniidae were selected using a Nikon SMZ-10 stereomicroscope (Nikon Co., Tokyo, Japan) at 6.6–40× magnification.

### 2.3. Staining and Morphological Identification

Morphological analyses were conducted only on cestodes with a scolex and a complete strobila, including gravid proglottids. The scolex (with adjacent immature segments), as well as several mature and gravid proglottids, were tried to stain using acetic carmine [51], with slight modifications. Briefly, specimens were rinsed with phosphate-buffered saline (PBS) and flattened between two glass slides for at least 24 h. They were then stained in carmine for 24 h, destained in 2% hydrochloric alcohol, dehydrated through an ethanol series (70%, 80%, 90%, 96%, 100%), cleared in xylene, and mounted on slides using Canada balsam. Once the mounting was set, morphological characteristics (number, size, and arrangement of rostellar hooks; morphology of mature and gravid segments) were examined in 20 individuals of each species (confirmed after genetic analysis) using identification keys [10,13,21,52] under a MOTIC BA210 microscope (Xiamen, China) with 4×–40× objectives.

### 2.4. DNA Extraction

For genetic analysis, proglottids were taken from all collected specimens, whether the strobila was complete or not. Proglottids were digested in 200 µL of TE buffer (100 µM Tris, 1 mM EDTA, pH 8.0), 200 µL of 10% SDS, and 15 µL of Proteinase K (1 µg/µL), incubated overnight at 70 °C in a Thermomixer Compact (Eppendorf, AG, Hamburg, Germany) with shaking. DNA was extracted using the phenol–chloroform method described 150 by Sambrook and Russell [53]. Total DNA was recovered in 100 µL of Milli-Q water 151 and stored frozen at −20 °C until use.

### 2.5. Molecular Identification of Hydatigera Species

A multiplex PCR assay was developed to identify *H. kamiyai* and *Hydatigera* sp. Based on distinct banding patterns. For optimization, 15 individuals were randomly selected and their species identified by PCR amplification and sequencing of a mitochondrial cytochrome c oxidase subunit 1 (cox1) fragment, using primers JB3 (5′-TTTTTTGGGCATCCTGAGGTTTAT) and JB4.5 (5′-TAAAGAAAGAACATAATGAAAATG) [54]. Reactions were made in 25 µL containing 5 µL of template DNA and 2 µL of 5 pmol/µL solution of each primer, using the PuReTaq Ready-To-Go PCR Beads kit (Merck KGaA, Darmstadt, Germany). Amplifications were conducted in a Mastercycler Gradient thermal cycler (Eppendorf AG, Hamburg, Germany) under the following conditions: initial denaturation at 94 °C for 10 min; 30 cycles of 94 °C for 1 min, 52 °C for 1 min, 72 °C for 1 min; and a final extension at 72 °C for 5 min. Amplified products were resolved on 1% agarose gels stained with Pronasafe (Condalab, Torrejón de Ardoz, Spain) and visualized under UV light using a Syngene transilluminator (NuGenius; Syngene, Cambridge, UK). PCR products were purified with the QIAquick PCR Purification Kit (QIAGEN, Hilden, Germany), and sequenced at the Genomics Unit of the Complutense University of Madrid using the JB3 primer on an AbiPrism 3730XL sequencer (Applied Biosystems, now Thermo Fisher Scientific, Waltham, MA, USA). Sequences were analyzed with ChromasPro v2.1.10.1 (Technelysium Pty Ltd., South Brisbane, Australia) and compared against sequences in GenBank/EMBL/DDBJ using the blastn algorithm on the NCBI website (https://blast.ncbi.nlm.nih.gov/Blast.cgi) (last accessed: 12 March 2025).

Once the species identity of each of the 15 selected samples was confirmed through molecular analysis, a multiplex PCR was optimized using these reference samples. Each reaction simultaneously amplified two distant regions of mitochondrial DNA: the cox1 gene, used as an internal control, and a diagnostic fragment including part of the cytochrome b gene (cytb, ~618 bp, for *Hydatigera* sp.), or encompassing the consecutive cytb-NADH dehydrogenase subunit 4 (nad4) genes (~1063 bp, for *H. kamiyai*). Based on complete mitochondrial DNA sequences of *Hydatigera* spp. available in GenBank (Table 2), a common forward primer for both species (HD; 5′-TATTACTGGTGATACATTAATGCGTG) and two species-specific reverse primers were designed: one for *H. kamiyai* (HKAR; 5′-AARTAAAAACGTACCCAACTAGACAG) and one for *Hydatigera* sp. (HSR; 5′-ATTAATCTTATCATAACGACAACTAATAATCC) (all primers’ solutions at 5 pmol/µL) (Table 3). The previously mentioned primers JB3 and JB4.5 were included in the reaction mix and used to amplify the cox1 fragment, serving as control of the reaction. Primer specificity was validated in silico using the NCBI Primer-BLAST tool on the NCBI website (https://www.ncbi.nlm.nih.gov/tools/primer-blast/index.cgi) (last accessed: 12 March 2025).

As a negative control (to ensure absence of nonspecific amplification of diagnostic fragments), DNA from *Hydatigera parva*, obtained from adult cestodes recovered during necropsy of a genet (*Genetta genetta*), was included in the assay.

Following optimization, all collected cestode individuals were analyzed using the multiplex PCR. Reactions were performed in a final volume of 25 µL, containing 3 µL of template DNA, 2 µL each of the common primers JB3, JB4.5, and HD, and 1.5 µL of each species-specific primer (HKAR and HSR), all at a final concentration of 5 µM. Thermal cycling conditions were: initial denaturation at 94 °C for 10 min; 30 cycles of 94 °C for 1 min, 52 °C for 1 min, and 72 °C for 2 min; followed by a final extension at 72 °C for 10 min.

Under these conditions, HD-JB4.5 amplification (~6660 positions) is not feasible because of insufficient time to complete the amplification. Therefore, each positive sample should display a control band of 450 bp (cox1), along with a species-specific band of either 620 bp cytb (*Hydatigera* sp.) or 1060 bp nad4 (*H. kamiyai*). In the negative control, only the 450 bp control band should be present. If no species-specific band is observed, the PCR product (then containing only the cox1 band) were purified and sequenced as previously described to confirm species identity.

## 3. Results

The wildcats analyzed were primarily collected in Castilla y León (16 from Burgos, 1 from Salamanca, 4 from Soria and 1 from Valladolid), while 4 individuals originated from Castilla-La Mancha (2 from Toledo, 1 from Guadalajara, and 1 from Ciudad Real) (Figure 1). A total of 73.1% (19/26) of the animals were infected with *Hydatigera* spp., including two individuals that showed coinfections with *Hydatigera* and *Joyeuxiella* (family Dilepididae). One animal (3.8%) was infected exclusively with *Joyeuxiella* sp.

A total of 240 *Hydatigera* spp. specimens were recovered, with parasite burden ranging from 4 to 36 cestodes per individual (Table 4 and Figure 1). The infected wildcats included 14 males and 5 females, and *Hydatigera* infection was detected in animals from all provinces except Ciudad Real, where the only sampled individual was infected solely with *Joyeuxiella*.

Morphometric data were obtained from specimens of *H. kamiyai* and *Hydatigera* sp. that preserved the scolex and a complete strobila. Species identification was confirmed by molecular analysis of immature proglottids prior to morphological comparison. Morphological data on the mature proglottids were not obtained because almost all of them were partially degenerated and internal reproductive structures such as testes, cirrus sac, and ovary were not clearly defined. The measurements of the main morphological structures, including scolex width, sucker dimensions, rostellar diameter, and hook morphology (large and small hooks), as well as the number of uterine branches (Figure 2), are summarized in Table 5. The size range of the different parameters overlapped for both species.

Sequences of the mitochondrial cox1 marker from the initial 15 cestodes analyzed corresponded to *H. kamiyai* in 9 individuals and to *Hydatigera* sp. in 6, with sequence similarities of 99–100% compared to available GenBank records. The *H. kamiyai* sequences displayed minor variability, with three haplotypes identified, whereas all *Hydatigera* sp. sequences were identical. The *H. parva* sequence showed 100% similarity with GenBank sequence NC021141, obtained from a cysticercus in a wood mouse (*Apodemus sylvaticus*) from northwestern Spain [11]. The sequences of *H. parva*, and the haplotypes of *H. kamiyai* and *Hydatigera* sp. were submitted to the GenBank/EMBL/DDBJ databases under accession numbers PV973980–PV973984.

The multiplex PCR successfully confirmed the identification of all 15 sequenced samples (Figure 3). When applied to the remaining cestodes, the multiplex PCR identified 128 out of 240 specimens (53.3%) as *H. kamiyai* and 112 (46.7%) as *Hydatigera* sp. Mixed infections were observed in 12 animals (60%), while single infections by *H. kamiyai* were found in 3 individuals (20.0%), and by *Hydatigera* sp. in 4 individuals (20.0%) (Table 4).

## 4. Discussion

This study presents, for the first time, data on the species-level identification of cryptic species within the *Hydatigera taeniaeformis* complex in wildcats in Spain. In this country, *H. taeniaeformis s.s* has previously been reported in intermediate hosts [21], and adult *H. taeniaeformis* s.l. has been recorded (Table 1), but this is the first study to identify both *H. kamiyai* and the unnamed European *Hydatigera* sp. in definitive hosts in Spain. *Hydatigera kamiyai* is mainly distributed across Europe and parts of Asia, whereas *H. taeniaeformis* s.s. shows a much broader, nearly cosmopolitan distribution, occurring also in Asia, Africa, Oceania, and the Americas (Table 1). However, European *Hydatigera* sp. has so far been reported only in Italy and France (and now in Spain).

Some internal structures (e.g., testes, ovaries, cirrus sac) appeared partially degenerated, likely due to the hosts’ post-mortem condition and freezing prior to necropsy, although more robust features such as the scolex and uterine branches were well preserved and could be measured accurately. The morphological characteristics of these structures did not allow differencing the adult stages of *H. kamiyai* and the European *Hydatigera* sp., a result which is in accordance with previous studies that stated the three species comprising the *H. taeniaeformis* complex exhibit very similar morphology [21]. Our data with *Hydatigera* sp. and *H. kamiyai* overlapped in all cases between them and with data published for *H. kamiyai* and *H. taeniaeformis* (Table 5). Although variations have been described in the number of proglottids, average number of rostellar hooks, orientation of small hooks, length of the cirrus sac, and number and position of testes [25,55], their morphological differentiation remains extremely difficult. Some authors argue that reliable distinction is only possible based on the measurements of rostellar hooks [21]. However, identification of the three species is achievable using molecular analyses, primarily based on the cox1 gene and, to a lesser extent, other molecular markers (mitochondrial 12S rRNA, NADH, nuclear 28S rRNA) [16,21,24,25,36,37]. According to the results of the present study, additional mitochondrial genes such as cytb and nad4 can also be used for the rapid identification of *H. kamiyai* and *Hydatigera* sp. The multiplex PCR developed in this research allows for the rapid and reliable processing of large sample sets and enables clear discrimination between the two species. Al-Sabi and coworkers [56] developed a multiplex PCR for cestode larval identification, although their method only distinguished *H. taeniaeformis* s.l. from other *Taenia* and *Versteria* species. A limitation of the current study is the lack of *H. taeniaeformis* s.s. samples, which prevented optimization of the multiplex PCR for differentiation among all three species within the *H. taeniaeformis* complex. To overcome this problem, a control band (corresponding to the partial amplification of the cox1 gene) was included to detect when the organism did not correspond to neither of the two species for which specific primers were used. In our opinion, it is necessary to include such controls in analysis based on presence/absence of bands after DNA PCR amplification using species-specific primers, both to detect potential new species or variations of previously described ones that would affect primers’ binding. This design allowed identification of the individual cestodes without needing sequencing, this speeding the identification and lowering analytical costs (amplicon purification and sequencing); sequencing would be limited to the cases where no specific bands were observed. This system is only fully valid to the analysis of separate, individual organisms; if a mix of individuals is analysed (for example, eggs or detached proglottids from a faecal sample), the presence of the species-specific bands does not exclude the existence of cestodes for which no specific primers were included in the analysis. Having this limitation in mind (detection will be limited to the species for which specific primers are used), our multiplex PCR system, as well as future developments including specific bands for other species, can be applied to faecal samples for epidemiological studies and to detect mixed infections.

In other studies involving molecular analyses of *Hydatigera* spp., species-level identification has typically been performed on only a small number of specimens (Table 1) leaving the actual distribution, prevalence, and frequency of co-infections largely un known.

*Hydatigera taeniaeformis* s.l. is the most frequently detected cestode in both domestic and wild felids, including wildcats [8,9], with prevalence values that in some cases exceed 50% [57,58,59,60] (Table 6). The high prevalence in wildcats may be associated with the importance of rodents in their diet [7], which serve as intermediate hosts [9]; to the best of our knowledge, there are no records of *Hydatigera* metacestodes in lagomorphs. The prevalence of taeniid cysticerci (including *H. taeniaeformis* s.l. and *H. parva*) in rodents in Spain ranges between 0.39–32.14% [61,62,63]. In the necropsies conducted in this study, all wildcats had mice in their stomachs, with 2 to 7 individuals per cat. This high predation rate increases the likelihood of infection. It remains to be determined whether the same rodent species can be hosts to both *H. kamiyai* and *Hydatigera* sp., which would require targeted investigation. According to available data, *H. taeniaeformis* s.s. has only been found in murid rodents, while *H. kamiyai* has been identified in murids, cricetids, soricids, and nesomyids (Table 1).

The prevalence of the species of the *H. taeniaeformis* complex detected in wildcats in this study (78%) is higher than that reported in other felid studies (Table 6). Since those previous studies were based solely on morphological identification, which prevents species-level assignment, it is not possible to establish the correct species identified in them. Molecular analyses have confirmed the presence of all three species of the *H. taeniaeformis* complex in Europe (Table 1), but these studies have been focused on species identification rather than epidemiology, and thus lack data on species-specific prevalence.

The parasite burden observed in wildcats ranged from 4 to 36 *Hydatigera* spp. cestodes per individual, with *H. kamiyai* showing the widest range (1 to 16 cestodes per host). This finding is consistent with previous reports, in which infection intensities ranged from 1 to 79 adult cestodes in felids (Table 4). However, inter-study comparisons at the species level are not reliable, as previous identifications have relied exclusively on morphological criteria.

## 5. Conclusions

This study reports for the first time the presence of *H. kamiyai* and European *Hydatigera* sp. in wildcats from Spain, expanding current knowledge of the distribution of cryptic species within the *H. taeniaeformis* complex. Considering the results of Lavikainen et al. [21] and the present results, the three species in the *H. taeniaeformis* complex are present in Spain. The multiplex PCR developed herein proved to be a rapid and reliable tool for species identification, overcoming the limitations of morphological analysis. Nonetheless, further studies including *H. taeniaeformis* s.s. samples are required to enable comprehensive molecular identification of all species within the *H. taeniaeformis* s.l. complex. The high prevalence and infection intensity observed highlight the epidemiological significance of these cestodes and the potential role of rodents as key intermediate hosts in their transmission cycle. These findings underscore the importance of implementing large-scale molecular approaches to clarify the true distribution and frequency of these species across Europe.

## Figures and Tables

**Figure 1 animals-15-03340-f001:**
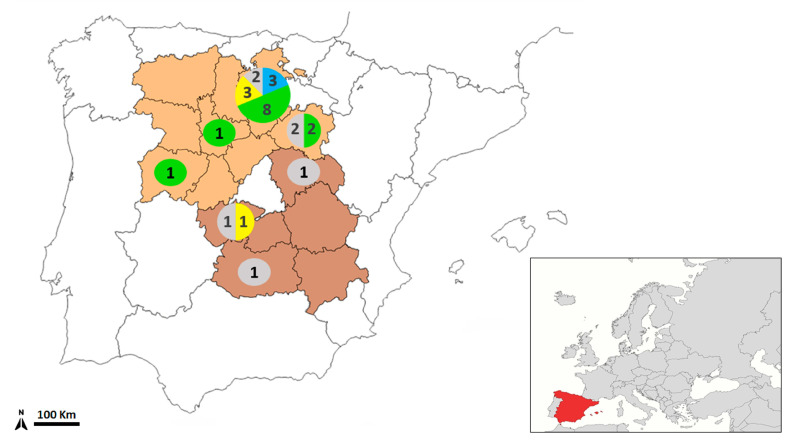
Geographic distribution of the *Hydatigera* species found in the analyzed European wildcats (*Felis silvestris*) in central Spain (country location shown in the inset, in red). The numbers in the circles indicate the number of wildcats non-infected/infected with one or more *Hydatigera* species according to the following color key: Grey: *Hydatigera* spp.-negative; Blue: *H. kamiyai*-positive; Yellow: *Hydatigera* sp.-positive; Green: Mix infections (*Hydatigera* sp. + *H. kamiyai*); Orange backgroupd: Castilla y León Autonomous Community; Dark Brown background: Castilla-La Mancha Autonomous Community.

**Figure 2 animals-15-03340-f002:**
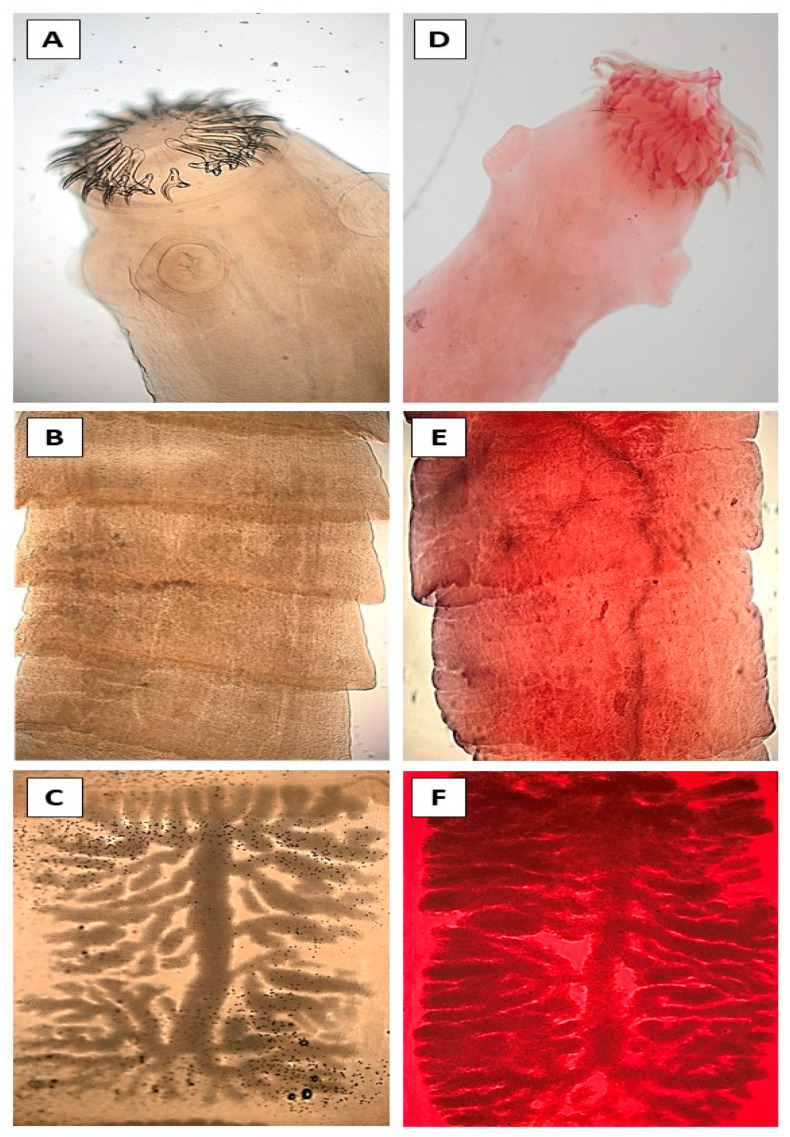
Scolex, mature and gravid proglottids of *Hydatigera* sp. (**A**–**C**) and *H. kamiyai* (**D**–**F**) and obtained from wildcats.

**Figure 3 animals-15-03340-f003:**
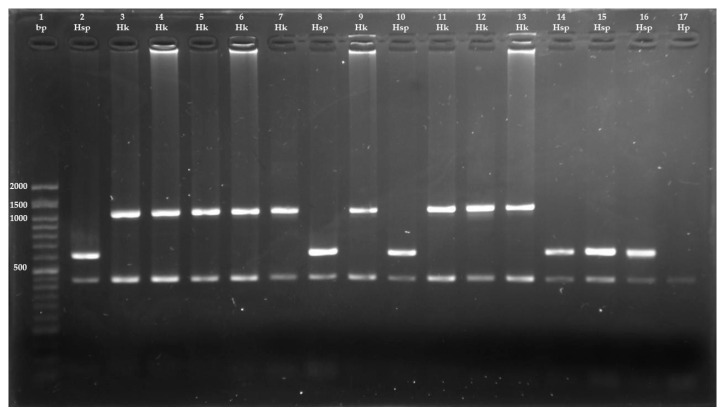
PCR multiplex band pattern on 1% agarose gel electrophoresis. From left to right: lane 1, Ladder (bp); lanes 2, 8, 10, 14, 15 and 16, *Hydatigera* sp. (Hsp); lanes 3, 4, 5, 6, 7, 9, 11, 12 and 13, *Hydatigera kamiyai* (Hk); lane 17, *Hydatigera parva* (Hp).

**Table 2 animals-15-03340-t002:** GenBank sequences utilized for multiplex PCR primer design.

*Hydatigera* Species	Stage	Host Species	GenBank Accession No.
*H. taeniaeformis* s.s.	Adult	*Felis catus*	FJ597547
	Adult	*Prionailurus bengalensis*	ON055368
	Adult	Not indicated	JQ663994
*H. kamiyai*	Adult	Not indicated	NC037071
	Adult	Not indicated	PP104554
	Adult	*Felis catus*	LC008533
*Hydatigera* sp.	Larva	*Eospolax fontanierii*	NC061206
	Larva	*Eospolax fontanierii*	MW808981
*H. parva*	Larva	Not indicated	NC021141
*H. krepkogorski*	Larva	Not indicated	NC021142

**Table 3 animals-15-03340-t003:** Complete mitochondrial sequences retrieved from GenBank utilized for multiplex PCR primer design. Primer location is shown in Appendix A.

	Primer	Sequence (5′-3′)	Expected Amplicon Size (bp)	Primer Location in mtDNA
Cestodes	JB3	TTTTTTGGGCATCCTGAGGTTTAT	450	cox1
	JB4.5	TAAAGAAAGAACATAATGAAAATG		
*H. kamiyai*	HD	TATTACTGGTGATACATTAATGCGTG	1063	cytb nad4
	HKAR	AARTAAAAACGTACCCAACTAGACAG		
*Hydatigera* sp.	HD	TATTACTGGTGATACATTAATGCGTG	618	cytb
	HSR	ATTAATCTTATCATAACGACAACTAATAATCC		

**Table 4 animals-15-03340-t004:** Prevalence of *Hydatigera kamiyai* and *Hydatigera* sp. in wildcats (*Felis silvestris*) from different locations in Spain.

Wildcat ID	Sex	Location	*Hydatigera kamiyai*	*Hydatigera* sp.	Total
147	H	Burgos	-	4	4
180	M	Burgos	11	6	17
198	M	Burgos	11	-	11
199	M	Burgos	6	4	10
211	M	Burgos	7	-	7
212	M	Burgos	8	-	8
236	H	Burgos	2	4	6
237	H	Soria	1	5	6
262	M	Burgos	19	1	20
263	M	Salamanca	11	4	15
268	M	Burgos	11	11	22
272	M	Valladolid	26	10	36
275	M	Burgos	3	7	10
317	M	Burgos	-	30	30
365	M	Burgos	-	13	13
366	M	Soria	2	3	5
367	H	Burgos	7	3	10
376	H	Burgos	3	2	5
412	M	Toledo	-	5	5
Total	5 H/14 M		128	112	240

Abbreviations: F = Female; M = Male.

**Table 5 animals-15-03340-t005:** Comparative morphometric measurements of *Hydatigera kamiyai* and *Hydatigera* sp. from wildcats in Spain, with reference data for *H. kamiyai* and *H. taeniaeformis* sensu stricto [21]. Values are given as mean ± standard deviation (range), in µm.

		Adults of Our Study (Mean)		Reference Values	
		*H. kamiyai*	*Hydatigera* sp.	*H. kamiyai*	*H. taeniaeformis* s.s.
Scolex width		1270 ± 200	1470 ± 100	1960 ± 200	1300 ± 125
		(1100–1580)	(1420–1550)	(1770–2170)	(1190–1440)
Rostellum diameter		791 ± 82.7	846 ± 21	824 ± 89.5	736 ± 38
		(700–902)	(817–863)	(731–910)	(703–779)
Number of hooks		32 ± 3.2	35 ± 0.9	33 ± 5	38 ± 3
		(28–36)	(34–36)	(30–40)	(36–42)
Length of large hooks		421 ± 29	409 ± 25	426 ± 30	429 ± 37
		(380–458)	(375–434)	(396–456)	(393–467)
Length of small hooks		269 ± 18	254 ± 14	253 ± 31	266 ± 16
		(266–286)	(247–273)	(213–275)	(249–281)
	TL	421 ± 29	409 ± 25	426 ± 30	425 ± 37
		(380–458)	(375–434)	(396–456)	(393–467)
	TW	169 ± 22	167 ± 18	162 ± 10.5	181 ± 12
Large		(145–205)	(142–184)	(150–171)	(170–194)
hooks	BL	280 ± 41	293 ± 9	265 ± 14	286 ± 29
		(224–322)	(281–302)	(249–277)	(256–314)
	AL	191 ± 18	178 ± 27	192 ± 15.5	202 ± 8
		(161–208)	(140–203)	(179–210)	(193–209)
	GL	71 ± 16	78 ± 15	75 ± 3.5	83 ± 11.5
		(54–88)	(56–88)	(71–78)	(72–95)
	GW	66 ± 7	76 ± 8	62 ± 4	68 ± 13
		(55–75)	(67–86)	(58–66)	(59–85)
	BC	40 ± 4	41 ± 5	37 ± 5.5	41 ± 7
		(37–43)	(39–47)	(32–43)	(35–49)
	HW	53 ± 15	51 ± 7	48 ± 6.5	64 ± 12.5
		(32–74)	(41–56)	(42–55)	(53–78)
	TL	269 ± 18	254 ± 14	253 ± 31	266 ± 16
		(266–286)	(247–273)	(213–275)	(249–281)
	TW	122 ± 8	130 ± 8	114 ± 4	123 ± 13
Small		(114–132)	(121–140)	(110–118)	(111–137)
hooks	BL	155 ± 18	156 ± 4	126 ± 22	150 ± 7
		(140–185)	(150–160)	(111–155)	(145–159)
	AL	142 ± 13	145 ± 1	141 ± 8.5	154 ± 10
		(124–161)	(144–146)	(131–148)	(146–166)
	GL	55 ± 6	61 ± 1	55 ± 6	55 ± 6
		(47–64)	(59–62)	(50–62)	(48–60)
	GW	56 ± 5	51 ± 11	44 ± 11	50 ± 11
		(49–62)	(39–65)	(35–57)	(40–62)
	BC	32 ± 6	32 ± 2	27 ± 7	38 ± 6
		(23–38)	(29–34)	(20–34)	(32–44)
	HW	34 ± 2	33 ± 6	31 ± 5	34 ± 5.5
		(33–37)	(25–40)	(25–35)	(29–40)
Sucker size (height × width)		401 ± 52 × 350 ± 96 (320–460) × (246–456)	381 ± 14 × 349 ± 38 (365–400) × (295–378)	445 ± 57 × 399 ± 65 (396–510) × (333–463)	300 ± 16.5 × 248 ± 20 (288–321) × (228–268)
Number of uterine		9 ± 1.5	9 ± 0.94	8 ± 2.5	9 ± 3.5
branches (unilateral)		(8–11)	(8–10)	(6–11)	(5–12)

Hook parameters following Lavikainen et al. [21]. TL: Total length; TW: Total width; BL: Basal length; AL: Apical length; GL: Guard length; GW: Guard width; BC: Blade curvature; HW: Handle width.

**Table 6 animals-15-03340-t006:** Epidemiological data of *Taenia taeniaeformis*/*Hydatigera taeniaeformis* s.l. across Felidae species.

Host Species *	Prevalence	Origin	Mean Intensity (Range)	Reference
*Lynx pardinus*	2/8 (25%)	Spain	1.50 (1–2)	[26]
*Lynx lynx*	1/37 (3%)	Estonia	1	[64]
*Felis silvestris*	8/15 (53%)	Germany	8 (2–20)	[57]
	17/23 (73.9%)	Greece	-	[60]
	7.7%	Scotland	-	[65]
	21/27 (78%)	Spain	12.6 (1–30)	This study
*Felis catus*	1/146 (0.68%)	Brazil	1	[66]
	14/358 (4%)	Mexico	3	[67]
	20/51 (39%)	Egypt	-	[68]
	370/488 (75.8%)	Qatar	-	[58]
	484/658 (73.6%)	Qatar	33.3	[59]
	40/240 (16.7%)	United Arab Emirates	4 (1–79)	[69]
	3/25 (12%)	Irak	-	[70]
	17/113 (15%)	Iran	0.35	[71]
	1/50 (2%)	Iran	-	[72]
	13/114 (12.3%)	Iran	-	[73]
	36/99 (36.4%)	Denmark	8.1 (1–57)	[74]
	5/162 (3.1%)	Portugal	(1–5)	[75]
	11/414 (2.7%)	Romania	-	[76]
	36/48 (75%)	Spain	-	[77]
	5/58 (8.6%)	Spain	-	[78]
*Prionailurus bengalensis*	1/1 (100%)	China	1	[79]

* Species named as *Felis silvestris catus* and *Felis silvestris silvestris* have been included as *Felis catus* or *F. silvestris* following the classification of Kitchener and coworkers (46) (Kitchener, 2017 [79]).

## Data Availability

All new data are presented in this study; data sharing is not aplicable to this article.

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
