# Peer review of "Identification of Hydatigera Species in Wildcats (Felis silvestris) from Central Spain"

_animals, 2025, doi:10.3390/ani15223340_

Round 1
Reviewer 1 Report
Comments and Suggestions for Authors
Dear Authors,
Your idea for distinguishing Hydatigera species is commendable; this is the first molecular study involving this host species and this genus in Spain. However, I would like to highlight key issues I have identified.
In general:
To summarize the most important points:
- The genus Hydatigera currently includes four valid species: kamiyai, H. taeniaeformis s.s., H. krepkogorski, and H. parva.
- Galimberti et al. (2012) published 61 sequences of Hydatigera tapeworms (from wild and domestic cats in Italy), identifying them as “Lineage 1.”
- Lavikainen et al. (2016) re-analyzed these sequences and concluded that they represent a potential new species, Hydatigera sp., characteristic of the Mediterranean region, further supported by an additional record from France (host:domestic cat) (GenBank LC008533, complete mt genome).
- Conclusion: within the H. taeniaeformis s.l. complex, three entities can be distinguished: H. taeniaeformis s.s., H. kamiyai, and Hydatigera sp.. The latter lacks sufficient morphological criteria and has so far only been suggested as a potential sister species of H. kamiyai.
- Wu et al. (2022) described a new Hydatigera sp. from rodents in China (GenBank NC061206 / MW808981 – representing the same sample). The authors do not discuss or link this finding to the European Hydatigera sp. (France, Italy).
- Rao et al. (2025) described yet another Hydatigera sp. in China, again independently from the previous European or Chinese findings (Galimberti 2012; Lavikainen 2016; Wu 2022). They claim to have identified a potential fifth species of the genus, but provide only the 18S rDNA sequence; the 12 PCG sequences mentioned in the paper are not available, which prevents meaningful comparison.
Attempt to clarify the issue
- The genetic distance between the French sequence (Lavikainen 2016) and the Chinese sequence (Wu 2022) is approximately 30%, strongly indicating distinct taxa. I calculated this value myself using complete mitochondrial genomes from GenBank. Although not reported in published literature, this figure illustrates the taxonomic confusion, which appears to have also affected the authors of the manuscript under review.
Major concerns with the present manuscript
- You state that your multiplex PCR primers (to distinguish H. kamiyai and Hydatigera sp.) were designed using sequences NC061206 and MW808981. However, both IDs correspond to the same Chinese sequence (Wu 2022). Why this sequence?
- It would have been more appropriate to use the French sequence (LC008533), particularly because your study was conducted in Spain and targets local populations of Hydatigera.
- Although you did include this French sequence (LC008533) in your primer design table, you incorrectly referred to it as H. kamiyai instead of Hydatigera sp.
- Reliance on the Chinese sequence may lead to misidentification and undermines the validity of your PCR analysis.
- It gives the impression that your manuscript attempts to develop a multiplex PCR for distinguishing a Hydatigera sp. lineage that has not yet been formally defined as a species, while it is not entirely clear which Hydatigera sp. is being targeted, nor has the taxonomy and phylogeny of this genus been adequately analyzed.
- According to my verification, the universal forward primer HD (designed in this study to amplify both H. kamiyai and Hydatigera sp.) binds correctly to NC037071 (H. kamiyai) and LC008533 (Hydatigera sp., France). The forward primer HKAR (for H. kamiyai) also performs as expected, but the forward primer HSR (claimed to be specific for Hydatigera sp.) does not match either the French or the Chinese sequence—it binds only partially, raising concerns about reproducibility.
- Nonetheless, I do observe bands separating at different base pairs, which appear to distinguish the species; I would appreciate it if you could dispel my doubt and clarify this.
- It also remains unclear why you focused solely on distinguishing H. kamiyai and Hydatigera sp., while excluding H. taeniaeformis s.s. from your multiplex PCR design.
- You used primers to amplify a fragment of the Hydatigera mitochondrial cytochrome c oxidase subunit 1 (cox1) gene, which you designated as COD (5′-TTTTTTGGGCATCCTGAGGTTTA) and COR (5′-TAAAGAAAGAACATAATGAAAATG). However, in the relevant literature on the genus Hydatigera, the same primers are consistently referred to as JB3 and JB45, with proper citation to the original design (Bowles et al., 1992). Luzón et al. (2010) used these primers in their study and assigned them local names CO1D and CO1R, but they did not design them. In your manuscript, although you cite Luzón et al. (2010), you incorrectly imply that these primers were originally theirs, additionally renaming them as COD/COR, and the sequences you provide are incorrect (missing one nucleotide at the 3′ end). Further, you indicate that after amplification with these primers, you performed sequencing and that the sequences of H. parva, H. kamiyai, and Hydatigera sp. were submitted to the GenBank/EMBL/DDBJ databases under accession numbers PV973980–PV973984. Based on this information, it appears that five sequences were deposited. It would be interesting and important to verify these sequences; however, at the time of review, they do not appear to be publicly accessible.
- The gel image presented in Figure 2 is not sufficiently clear. The base pair ruler is not well visible, although the bands corresponding to the two different species are recognizable. Since you aim to demonstrate a new method for species differentiation, the quality of the gel image is crucial for supporting your argument. Additionally, it would be helpful if the lanes were clearly labeled so that the reader would not have to count them manually.
- Another point of concern relates to the sample preparation: you state that you performed a separate DNA isolation for each Hydatigera individual (i.e., all 240 individuals underwent separate isolation and separate amplification), with each band on the gel corresponding to a single species. However, it remains unclear what would happen if DNA from both species were present in the same sample. Would two bands appear in that case? This is important for assessing the specificity of your method and its applicability under real-world conditions.
Introduction:
In the Introduction, it is necessary to place more emphasis on explaining the genus Hydatigera and its species. The work of Galimberti et al. (2012) should be described in more detail, as they initiated the discussion on Hydatigera sp., followed by the record from France (Lavikainen et al., 2016). It is important to highlight the significance of this species, which is likely restricted to the Mediterranean region, and to note that, to my knowledge, you are the third group to study Hydatigera sp..This is particularly important because Hydatigera sp. has not yet been formally described as a species and very few morphological data are available. In this context, the Introduction states: “Therefore, the aims of this study were to morphologically and molecularly identify the Hydatigera species infecting European wildcats in Spain.” However, in the Results section, I do not see clear presentation of morphological characteristics, differences, or specific values observed. It is necessary to clarify exactly what morphological traits were identified and how these data support your conclusions.
Row 60–61: Check reference (10) – the title currently cites a plant name (Tordylium), whereas it likely refers to: Verster, A. (1969). A taxonomic revision of the genus Taenia Linnaeus, 1758 s. str. Onderstepoort J. Vet. Res., 36(1), 3–58. Additionally, in the section on Hydatigera krepkogorski (Schulz and Landa, 1934), Hydatigera parva (Baer, 1924), and Hydatigera taeniaeformis (Batsch, 1786), it would be helpful to cite more recent sources confirming the hosts of these species, rather than relying solely on these older references.
Row 65–66: The statement "The only available data on H. kamiyai and Hydatigera sp. derive mainly from isolates obtained from felids (14,16,17), and in some cases from rodents (18)" is not entirely accurate. A more comprehensive review of the cited studies is necessary. Additionally, I recommend consulting recent literature, such as the work by Martini et al. (2022), Al sabi et al (2015), Miljević et al (2023), Catalano et al (2019) an others…
Row 70-71 Reference: Lozano, J. (2017). Gato montés – Felis silvestris. Enciclopedia Virtual de los Vertebrados Españoles. CSIC, Museo Nacional de Ciencias Naturales, Madrid.
This encyclopedia provides general information on the biology, ecology, and habitat of the wildcat (Felis silvestris), including a note that wildcats can be infected with various parasites. However, it does not specify particular cestode species or provide prevalence data, meaning it cannot support the authors’ claim that “the prevalence of cestodes in wildcats in Spain is high.” In addition to this encyclopedia, the authors cite a conference abstract and a study over 30 years old. It would be useful to clarify on what data the claim of high prevalence is based. For the statement regarding high cestode prevalence in wildcats in Spain, empirical studies providing concrete data should be cited. If no such recent studies exist, it would be more appropriate to note, for example, that there is a single study from 1989 reporting cestode prevalence and that no new data have been published since, indicating a gap in the literature.
Discussion
Row 208-211 The statement: “In this country, H. kamiyai has previously been reported in intermediate hosts (16), and adult H. taeniaeformis s.l. has been recorded (Table 4), but this is the first study to identify both H. kamiyai and Hydatigera sp. in definitive hosts in Spain” is incorrect. In fact, the cited study (Lavikainen et al., 2016) detected (haplotip A12 -H. taeniaeformis s.s.) in intermediate hosts (Mus musculus) in Spain, not H. kamiyai. Interestingly, on the European continent, H. taeniaeformis s.s. has been reported in small mammals only in Spain and Belgium, whereas in all other regions, H. kamiyai is the species detected.
Additionally, there is no need to refer to Table 4 here, and it is not necessary at all. What exactly are you reporting in this table? – Hydatigera taeniaeformis s.l., which encompasses all species within this complex that may not have been precisely determined because molecular analysis was not used. Yet, you place your results into this table even though you have already clearly identified the specific species you have. Furthermore, the table lists random findings from around the world, which do not seem to have been of practical use even in the discussion.
The entire sentence structure and the reference to the table need to be revised.
Furthermore, you state: “In other parts of Europe, all three species of the H. taeniaeformis complex have been detected in definitive and/or intermediate hosts (Table 5).” However, the table lists countries from all over the world – from Australia, through Mexico and Brazil, to Europe. Why does the text not highlight that Hydatigera sp. has been detected only in Italy and France? It would also be important to clarify what is specific about the occurrence of H. taeniaeformis s.s. in Europe versus Asia. As it stands, the text gives the impression of a global distribution of all species, without emphasizing key geographic and taxonomic distinctions.
Dear authors, it gives the impression that there are significant gaps in the understanding of the taxonomic status of the genus Hydatigera and its constituent species. Please clarify which Hydatigera sp. you are studying, why it is important, and why it is relevant for your region. If you already have morphological results, please include them and highlight the significance of your study. Additionally, discuss the potential application of this method using carnivore feces as a sample – have you tried this approach? It is very important for me to know whether the method works if multiple Hydatigera species are present in a single sample. Furthermore, please explain how you exclude the possibility of H. taeniaeformis s.s. All of these points should be considered and addressed in your discussion.
I encourage you to address and clarify these points, as the development of a multiplex PCR capable of distinguishing Hydatigera spp. would represent a valuable and useful contribution to the field
Regards,
Author Response
Responses to Reviewer 1
Comment 1: Dear Authors, Your idea for distinguishing Hydatigera species is commendable; this is the first molecular study involving this host species and this genus in Spain. However, I would like to highlight key issues I have identified.
In general:
To summarize the most important points:
- The genus Hydatigera currently includes four valid species: kamiyai, H. taeniaeformis s.s., H. krepkogorski, and H. parva.
- Galimberti et al. (2012) published 61 sequences of Hydatigera tapeworms (from wild and domestic cats in Italy), identifying them as “Lineage 1.”
- Lavikainen et al. (2016) re-analyzed these sequences and concluded that they represent a potential new species, Hydatigera sp., characteristic of the Mediterranean region, further supported by an additional record from France (host:domestic cat) (GenBank LC008533, complete mt genome).
- Conclusion: within the H. taeniaeformis s.l. complex, three entities can be distinguished: H. taeniaeformis s.s., H. kamiyai, and Hydatigera sp.. The latter lacks sufficient morphological criteria and has so far only been suggested as a potential sister species of H. kamiyai.
- Wu et al. (2022) described a new Hydatigera sp. from rodents in China (GenBank NC061206 / MW808981 – representing the same sample). The authors do not discuss or link this finding to the European Hydatigera sp. (France, Italy).
- Rao et al. (2025) described yet another Hydatigera sp. in China, again independently from the previous European or Chinese findings (Galimberti 2012; Lavikainen 2016; Wu 2022). They claim to have identified a potential fifth species of the genus, but provide only the 18S rDNA sequence; the 12 PCG sequences mentioned in the paper are not available, which prevents meaningful comparison. Attempt to clarify the issue
- The genetic distance between the French sequence (Lavikainen 2016) and the Chinese sequence (Wu 2022) is approximately 30%, strongly indicating distinct taxa. I calculated this value myself using complete mitochondrial genomes from GenBank. Although not reported in published literature, this figure illustrates the taxonomic confusion, which appears to have also affected the authors of the manuscript under review.
Response 1: Thanks very much for this quick and accurate explanation of the taxonomic situation in the genus Hydatigera. As we will indicate below, we think we have considered correctly the present situation. It is true that we have not made comparisons of the European Hydatigera sp. with the Asian Hydatigera sp. (sp1 and sp2), because they are clearly different species but the Asian species have been identified from rodents (intermediate hosts) and the definitive host is at this moment unknown, and we are dealing with the findings in the definitive host (in our case, in wildcats, specifically in Spanish wildcats).
Comment 2. Major concerns with the present manuscript
You state that your multiplex PCR primers (to distinguish H. kamiyai and Hydatigera sp.) were designed using sequences NC061206 and MW808981. However, both IDs correspond to the same Chinese sequence (Wu 2022). Why this sequence?
It would have been more appropriate to use the French sequence (LC008533), particularly because your study was conducted in Spain and targets local populations of Hydatigera.
Although you did include this French sequence (LC008533) in your primer design table, you incorrectly referred to it as H. kamiyai instead of Hydatigera sp.
Reliance on the Chinese sequence may lead to misidentification and undermines the validity of your PCR analysis.
Response 2: The Rao et al. sequences were not available when started this study and designed the primers. Besides, as the reviewer indicates, the sequences of the PCG are not available and for this reason, they cannot be included in the alignment for designing the primers. For this reason, they have not been included in Table 1.
In order to identify the species, we first identified the species we could find in our sampling by analysing a subset of the cestodes (15 individuals). We identified two species: H. kamiyai and the European Hydatigera sp. For developing a quick method to differentiate between these two species, we aimed to designed primers that were specific for each of them. In our design, if amplification of the specific band were not produced (but it did the control band), then a different species would be present (i.e., in such case, it could be H. taeniaeformis s.s., H. parva, H. krepkogorski, the Asian sp1 or sp2, or even a new, different species). In such case, sequencing of the control band would have been done to identify the species, we would have revised the alignment and designed new primers for that additional species and re-adjusted the conditions for the multiplex PCR.
Considering this study design, the sequences we considered for primer design corresponded to H. taeniaeformis s.s. (FJ597547 and ON055368); H. kamiyai (JQ663994, NC037071 and PP104554), Hydatigera sp. (European LC008533, Asian NC061206 and MW808981), H. parva (NC021141) and H. krepkogorski (NC021142).
There can be a problem in reading previous Table 1 (now Table 2) due to text aligning: the first column (Hydatigera species) has the cells “centered vertically”, this leading to confusion about the sequences corresponding to each sequence. We have added horizontal lines to new Table 2 to clearly separate the sequences of each species.
Comment 2. It gives the impression that your manuscript attempts to develop a multiplex PCR for distinguishing a Hydatigera sp. lineage that has not yet been formally defined as a species, while it is not entirely clear which Hydatigera sp. is being targeted, nor has the taxonomy and phylogeny of this genus been adequately analyzed.
Response 2: As mentioned in the previous response, the primers we designed targeted the European Hydatigera sp. and H. kamiyai, which were the species we initially found in our animals. The intention was not to distinguish the European Hydatigera sp. from other species, but to distinguish all the species present in the sampled animals. In the present conditions, the multiplex PCR would detect if a given Hydatigera cestode is H. kamiyai, the European Hyatigera sp., or a different species (in this case, the control band, corresponding to the cox1 gene, should be sequenced to determine the species).
Comment 3. According to my verification, the universal forward primer HD (designed in this study to amplify both H. kamiyai and Hydatigera sp.) binds correctly to NC037071 (H. kamiyai) and LC008533 (Hydatigera sp., France). The forward primer HKAR (for H. kamiyai) also performs as expected, but the forward primer HSR (claimed to be specific for Hydatigera sp.) does not match either the French or the Chinese sequence—it binds only partially, raising concerns about reproducibility.
Nonetheless, I do observe bands separating at different base pairs, which appear to distinguish the species; I would appreciate it if you could dispel my doubt and clarify this.
Response 3. We have added a Supplementary file 1 containing the alignment of the sequences used for primer design, and the location of the primers. As it can be seen in the alignment, the primers corresponded to the specific sequences targeting each species. Primers whose name end in ‘D’ are direct/forward, while those ending in ‘R’ are reverse. Primer HD is forward and is located, in the alignment in sup. Fig. 1, at position ~1270; HSR and HKAR are reverse and are located at positions ~1850 and ~2300, respectively. We have corrected the sequence of primer HKAR, the 5’ end is AART; we erroneously copied, for these 4 positions, the complementary sequence, TTYA. The correct primer HKAR is 5’-AARTAAAAACGTACCCAACTAGACAG.
Comment 4. It also remains unclear why you focused solely on distinguishing H. kamiyai and Hydatigera sp., while excluding H. taeniaeformis s.s. from your multiplex PCR design.
Response 4. Our initial intention was to develop a multiplex PCR to distinguish between the three species in the H. taeniaeformis complex. However, we did not found H. taeniaeformis in our sampled animals. We have tried to obtain H. krepkogorski DNA, and we asked prof. Mitra Sharbatkhori, who recently (2018) analysed individuals of this species (doi: 10.30466/vrf.2018.33105), but he indicated that currently do not have available material from this species nor from H. taeniaeformis s.s. We also checked the possibility of obtaining H. taeniaeformis s.s. by contacting prof. Antti Lavikainen, but in this case, we had no response. If we would have found it at any moment when performing the analysis of all individuals, we would have revised the primers, added new ones for this species, and tested a revised version of the multiplex PCR.
Comment 5. You used primers to amplify a fragment of the Hydatigera mitochondrial cytochrome c oxidase subunit 1 (cox1) gene, which you designated as COD (5′ TTTTTTGGGCATCCTGAGGTTTA) and COR (5′ TAAAGAAAGAACATAATGAAAATG). However, in the relevant literature on the genus Hydatigera, the same primers are consistently referred to as JB3 and JB45, with proper citation to the original design (Bowles et al., 1992). Luzón et al. (2010) used these primers in their study and assigned them local names CO1D and CO1R, but they did not design them. In your manuscript, although you cite Luzón et al. (2010), you incorrectly imply that these primers were originally theirs, additionally renaming them as COD/COR, and the sequences you provide are incorrect (missing one nucleotide at the 3′ end). Further, you indicate that after amplification with these primers, you performed sequencing and that the sequences of H. parva, H. kamiyai, and Hydatigera sp. were submitted to the GenBank/EMBL/DDBJ databases under accession numbers PV973980–PV973984. Based on this information, it appears that five sequences were deposited. It would be interesting and important to verify these sequences; however, at the time of review, they do not appear to be publicly accessible.
Response 5. The reviewer is right, the original design of the primers corresponds to Bowles et al., not to Luzón et al. We have revised the manuscript to change the primer names: COD-> JB3, COR->JB4.5 and replaced the reference. We have corrected the sequence of primer JB3 in new table 3 (a previously missing ‘T’ was added in the 3’ end).
In relation to the sequences submitted to Genbank, it is a common procedure to keep them confidential (“embargoed”) for a given period of time, and they are made public once they are used in a publication or the embargoed period finished (what happens first).
Comment 6. The gel image presented in Figure 2 is not sufficiently clear. The base pair ruler is not well visible, although the bands corresponding to the two different species are recognizable. Since you aim to demonstrate a new method for species differentiation, the quality of the gel image is crucial for supporting your argument. Additionally, it would be helpful if the lanes were clearly labeled so that the reader would not have to count them manually.
Response 6. Previous Figure 2 (now Figure 3) has been replaced, and both the weight marker and lanes have been labelled.
Comment 7. Another point of concern relates to the sample preparation: you state that you performed a separate DNA isolation for each Hydatigera individual (i.e., all 240 individuals underwent separate isolation and separate amplification), with each band on the gel corresponding to a single species. However, it remains unclear what would happen if DNA from both species were present in the same sample. Would two bands appear in that case? This is important for assessing the specificity of your method and its applicability under real-world conditions.
Response 7. The reviewer is correct in that each individual cestode underwent separate DNA isolation and amplification. Then, as the individual corresponded to a single species, only the control (common) band and the specific band will appear. If the cestode under analysis is of a different species than H. kamiyai or the European Hydatigera sp., only the control band would be present (see new Figure 3, H. parva). A three-band amplification would be only possible if DNA of different species are simultaneously amplified, but we use single individuals (only those having the scolex were analyzed). A mixed pattern is only possible if tissues (say, proglottids) from individuals of different species are mixed and processed as a pool. In such case, the bands corresponding to each species would be present. To demonstrate this, we have made a new DNA extraction and PCR amplification using proglottids from two different individuals (one corresponding to H. kamiyai and the other to Hydatigera sp.). As you can see in the next image, the bands corresponding to each species is present in the mixed DNA. The first lane contains the molecular marker, lanes 2 and 3 (mix1, mix2) correspond to amplification of DNA from mixed proglottids, and lanes 4 and 5 correspond to amplification of DNA from H. kamiyai (Hk) and European Hydatigera sp. (Hsp). In all cases, the control band (partial cox1 gene) is present.
We could have done a fast screening by analyzing pools of proglottids from different individuals and check what specific bands were present: only the control band, this and an unique specific band, or the control band and both specific bands; in the first and latter case, separate analysis of the individuals in the initial pool should were made to identify each of them. However, in the second option (only one specific band were present) this would have overlooked the presence of a third species (no specific band for such unknown species would have appeared). For this reason, we did the analysis for each of the individuals separately.
Comment 8. Introduction:
In the Introduction, it is necessary to place more emphasis on explaining the genus Hydatigera and its species. The work of Galimberti et al. (2012) should be described in more detail, as they initiated the discussion on Hydatigera sp., followed by the record from France (Lavikainen et al., 2016). It is important to highlight the significance of this species, which is likely restricted to the Mediterranean region, and to note that, to my knowledge, you are the third group to study Hydatigera sp..This is particularly important because Hydatigera sp. has not yet been formally described as a species and very few morphological data are available. In this context, the Introduction states: “Therefore, the aims of this study were to morphologically and molecularly identify the Hydatigera species infecting European wildcats in Spain.” However, in the Results section, I do not see clear presentation of morphological characteristics, differences, or specific values observed. It is necessary to clarify exactly what morphological traits were identified and how these data support your conclusions.
We have modified the introduction to include a more detailed explanation of the taxonomy of the H. taeniaeformis complex (lines 62-80). In relation to the morphological analysis, we have added the results of our analysis in the Results section (lines 241-244) and made some comments in Discussion (lines 280-286).
Comment 9. Row 60–61: Check reference (10) – the title currently cites a plant name (Tordylium), whereas it likely refers to: Verster, A. (1969). A taxonomic revision of the genus Taenia Linnaeus, 1758 s. str. Onderstepoort J. Vet. Res., 36(1), 3 58. Additionally, in the section on Hydatigera krepkogorski (Schulz and Landa, 1934), Hydatigera parva (Baer, 1924), and Hydatigera taeniaeformis (Batsch, 1786), it would be helpful to cite more recent sources confirming the hosts of these species, rather than relying solely on these older references.
Response 9. Thanks very much for noting the error in the reference. It was a mistake when adding the reference using a reference manager. It has been corrected. We have also included two more recent cites on the hosts of Hydatigera species, but there are almost no new data on the definitive hosts of H. parva and H. krepkogorski (lines 60-62).
Comment 10. Row 65–66: The statement "The only available data on H. kamiyai and Hydatigera sp. derive mainly from isolates obtained from felids (14,16,17), and in some cases from rodents (18)" is not entirely accurate. A more comprehensive review of the cited studies is necessary. Additionally, I recommend consulting recent literature, such as the work by Martini et al. (2022), Al sabi et al (2015), Miljević et al (2023), Catalano et al (2019) an others.
We have added more recent cites on the hosts of Hydatigera species, but there are almost no data for other species than H. taeniaeformis. We have rewritten the paragraph in lines (62-80) in introduction.
Comment 11. Row 70-71 Reference: Lozano, J. (2017). Gato montés – Felis silvestris. Enciclopedia Virtual de los Vertebrados Españoles. CSIC, Museo Nacional de Ciencias Naturales, Madrid.
This encyclopedia provides general information on the biology, ecology, and habitat of the wildcat (Felis silvestris), including a note that wildcats can be infected with various parasites. However, it does not specify particular cestode species or provide prevalence data, meaning it cannot support the authors’ claim that “the prevalence of cestodes in wildcats in Spain is high.” In addition to this encyclopedia, the authors cite a conference abstract and a study over 30 years old. It would be useful to clarify on what data the claim of high prevalence is based. For the statement regarding high cestode prevalence in wildcats in Spain, empirical studies providing concrete data should be cited. If no such recent studies exist, it would be more appropriate to note, for example, that there is a single study from 1989 reporting cestode prevalence and that no new data have been published since, indicating a gap in the literature.
We have modified the last paragraph of introduction (lines 81-100) a follows: “There are almost no previous data on cestode infections in wildcats in Spain; the only data available (referred to H. taeniaeformis s.l.) are those obtained more than 30 years ago by Torres et al. (1989) in samples from all over Spain, who found a overall 60.3% prevalence; and those obtained between 2008-2015 by Gómez-Galindo et al. (2019) in samples collected in the south-east of Spain, who found a 36.8% prevalence. Other cestodes found in these studies included Taenia pisiformis, Joyeuxiella pasqualei, Diplopylidium nolleri and Mesocestoides spp. In both studies, identifications were based on morphological characters. To date, no studies have described the clinical manifestations of Hydatigera infection in European wildcats, and data in domestic cats are also scarce; in general, as with most intestinal cestodes in felids, infections by adult Hydatigera tapeworms are considered largely subclinical [8, 9]. However, a few isolated clinical cases have been reported in domestic cats, including an acute intestinal obstruction caused by (Taenia) Hydatigera taeniaeformis s.l., which required surgical removal of the tapeworms [28]. This highlights that, although rare, heavy infections may lead to clinical disease. From an epidemiological perspective, identifying Hydatigera species is important because they differ in their life cycles, intermediate hosts, and geographic distributions, thus providing insights into trophic relationships and potential transmission pathways between wild and domestic carnivores. There is an important gap in the literature on Hydatigera infections in Spanish wildcats, and the objective of this study is provide new, recent data on the presence and distribution of this cestode genus in the Spanish European wildcats.”
Comment 12. Discussion
Row 208-211 The statement: “In this country, H. kamiyai has previously been reported in intermediate hosts (16), and adult H. taeniaeformis s.l. has been recorded (Table 4), but this is the first study to identify both H. kamiyai and Hydatigera sp. in definitive hosts in Spain” is incorrect. In fact, the cited study (Lavikainen et al., 2016) detected (haplotip A12 -H. taeniaeformis s.s.) in intermediate hosts (Mus musculus) in Spain, not H. kamiyai. Interestingly, on the European continent, H. taeniaeformis s.s. has been reported in small mammals only in Spain and Belgium, whereas in all other regions, H. kamiyai is the species detected.
Response 12. Thank you very much for noting the error; effectively, the species should be H. taeniaeformis s.s. We have modified the paragraph to correct this error and added the following text (lines 265-272) to indicate that the species of the H. taeniaeformis complex are present in Europe (but still not all detected in a same country): “In this country, H. taeniaeformis s.s has previously been reported in intermediate hosts (16)(Lavikainen, 2016), and adult H. taeniaeformis s.l. has been recorded (Table 1), but this is the first study to identify both H. kamiyai and the unnamed European Hydatigera sp. in definitive hosts in Spain. Hydatigera kamiyai is mainly distributed across Europe and parts of Asia, whereas H. taeniaeformis s.s. shows a much broader, nearly cosmopolitan distribution, occurring also in Asia, Africa, Oceania, and the Americas (Table 1). However, European Hydatigera sp. has so far been reported only in Italy and France (and now in Spain).”
”
Comment 13. Additionally, there is no need to refer to Table 4 here, and it is not necessary at all. What exactly are you reporting in this table? – Hydatigera taeniaeformis s.l., which encompasses all species within this complex that may not have been precisely determined because molecular analysis was not used. Yet, you place your results into this table even though you have already clearly identified the specific species you have. Furthermore, the table lists random findings from around the world, which do not seem to have been of practical use even in the discussion.
Response 13. There are not previous data on the intensity of infection of the species in the Hydatigera taeniaeformis complex. For comparison purposes, and considering the sensu lato species, we added this table to be able to compare our results (intensity of infection) with those already published. We have included published articles from around the world in which data on intensity of infection were indicated. After revising the manuscript, we have kept this table and changed its number to Table 6.
Comment 14. Furthermore, you state: “In other parts of Europe, all three species of the H. taeniaeformis complex have been detected in definitive and/or intermediate hosts (Table 5).” However, the table lists countries from all over the world – from Australia, through Mexico and Brazil, to Europe. Why does the text not highlight that Hydatigera sp. has been detected only in Italy and France? It would also be important to clarify what is specific about the occurrence of H. taeniaeformis s.s. in Europe versus Asia. As it stands, the text gives the impression of a global distribution of all species, without emphasizing key geographic and taxonomic distinctions.
Response 14. Please see response 12.
Comment 15. Dear authors, it gives the impression that there are significant gaps in the understanding of the taxonomic status of the genus Hydatigera and its constituent species. Please clarify which Hydatigera sp. you are studying, why it is important, and why it is relevant for your region. If you already have morphological results, please include them and highlight the significance of your study. Additionally, discuss the potential application of this method using carnivore feces as a sample – have you tried this approach? It is very important for me to know whether the method works if multiple Hydatigera species are present in a single sample. Furthermore, please explain how you exclude the possibility of H. taeniaeformis s.s. All of these points should be considered and addressed in your discussion. I encourage you to address and clarify these points, as the development of a multiplex PCR capable of distinguishing Hydatigera spp. would represent a valuable and useful contribution to the field.
Response 15. We think we have adequately resolved the query about what Hydatigera sp. (the European one) was identified, in previous responses. We have modificated the text in discussion to clarify this and why it is relevant four our region (lines 271-272). Please see response 12. “
We have added the data from the morphological part of the study (lines 241-244). In relation to the two points about the genetic analyses (potential application to the analysis of feces, and Hydatigera spp. Identification), they are dealt with in the discussion section. We have added the following text at the end of the second paragraph (lines 295-314): “A limitation of the current study is the lack of H. taeniaeformis s.s. samples, which prevented optimization of the multiplex PCR for differentiation among all three species within the H. taeniaeformis complex. To overcome this problem, a control band (corresponding to the partial amplification of the cox1 gene) was included to detect when the organism did not correspond to neither of the two species for which specific primers were used. In our opinion, it is necessary to include such controls in analysis based on presence/absence of bands after DNA PCR amplification using species-specific primers, both to detect potential new species or variations of previously described ones that would affect primers’ binding. This design allowed identification of the individual cestodes without needing sequencing, this speeding the identification and lowering analytical costs (amplicon purification and sequencing); sequencing would be limited to the cases where no specific bands were observed. This system is only fully valid to the analysis of separate, individual organisms; if a mix of individuals is analysed (for example, eggs or detached proglottids from a faecal sample), the presence of the species-specific bands does not exclude the existence of cestodes for which no specific primers were included in the analysis. Having this limitation in mind (detection will be limited to the species for which specific primers are used), our multiplex PCR system, as well as future developments including specific bands for other species, can be applied to faecal samples for epidemiological studies and to detect mixed infections.”
Reviewer 2 Report
Comments and Suggestions for Authors
The article addresses the presence of Hydatigera in wildcats that were road-killed in Spain, along with the development of a multiplex PCR assay capable of distinguishing between two cryptic species of this genus that are morphologically indistinguishable. Overall, the study is well structured and the methods are clearly described. However, I recommend that some revisions be made to improve the quality of the manuscript:
In the Introduction, I believe the reason for studying this parasite in these wild hosts should be better substantiated. It would be important to clarify the potential impact in terms of animal and public health, as well as to explain why distinguishing between the different species is relevant. Providing this background information would help the reader better understand the significance of the study and make it more engaging.
In the Methodology, the technique used for morphological identification is described, but it is not mentioned at any point in the Results section, it appears that all identifications were carried out solely by molecular methods. Furthermore, since the authors frequently refer to these species as cryptic, it should be clarified why morphological identification was attempted in the first place.
Figure 1 (map) requires major improvement. It currently lacks a geographic scale, a north arrow, and a reference indicating the source of the map (program where the map was developed). I also suggest including a small inset map of Europe showing the location of Spain.
Figure 2 is not very intuitive for the reader. I recommend adding labels directly on the figure to indicate what each band or well represents.
In the Discussion, the authors emphasize the presence of this parasite in rodents. However, since the Introduction mentions that lagomorphs are also part of the wildcats’ diet, I missed seeing any discussion or information regarding the occurrence of this parasite in that host group.

Author Response
Responses to Reviewer 2
Comment 1: In terms of health, does these parasites affect the wildcats? Does it generate clinical symptoms? From a health perspective, do these parasites affect wildcats, and do they cause any clinical symptoms? From an epidemiological standpoint, why is it important to study these parasites, and more specifically, to distinguish among their species? Do different species produce distinct clinical manifestations? Providing this information to readers is essential to clarify and emphasize the significance of this study.
Response 1: We have added the following text at the end of introduction (lines 88-98): “To date, no studies have described the clinical manifestations of Hydatigera infection in European wildcats (Felis silvestris), and data in domestic cats (Felis catus) are also scarce. In general, as with most intestinal cestodes in felids, infections by adult Hydatigera tapeworms are considered largely subclinical. However, a few isolated clinical cases have been reported in domestic cats, including an acute intestinal obstruction caused by (Taenia) Hydatigera taeniaeformis, which required surgical removal of the tapeworms (Wilcox et al., 2009). This highlights that, although rare, heavy infections may lead to clinical disease. From an epidemiological perspective, identifying Hydatigera species is important because they differ in their life cycles, intermediate hosts, and geographic distributions, thus providing insights into trophic relationships and potential transmission pathways between wild and domestic carnivores.”
Comment 2: is there a reference that assures this estimation was close to the correct?
Response 2: We have added this paragraph to explain the estimation of age: “The classification of specimens as adults (over one year old) was based primarily on dental analysis, supplemented by body size and weight. Dental examination confirmed complete eruption of the permanent dentition. It is essential to note that the apical foramen of the canine root was closed and that the teeth showed incipient wear on the cusps of the canines and incisors. This level of dental wear, combined with evidence of a fully mature and closed tooth root, constitutes a non-invasive criterion established in the literature on wildcats for reliably classifying an individual in the >1 year age class [Oleinik, 2024; García-Perea and Baquero, 1999].
Comment 3: Insert the author name before the (26) citation
Response: We have modified the text as requested: “…..described by Sambrook and Russell (2006) [53]”
Comment 4: Remember to add an indentation at the beginning of each paragraph
Response 4: Done as requested.
Comment 5: The results of the morphological identification are missing, from what I've read it seems like the authors were based solely on the molecular assays.
In the methodology, the technique used for morphological identification is described, but it is not mentioned at any point in the results section. It appears that all identifications were carried out solely by molecular methods. Furthermore, since the authors frequently refer to these species as cryptic, it should be clarified why morphological identification was attempted in the first place.
Response 5: We have added the data of the morphological identification, this including a new table 5 and figure 2.
Respect to the second part of the comment, we think it is important to check the morphology of the isolates. It has been previously described that the species in the H. taeniaeformis complex were morphologically identical, but doing a comparative study (as the present one) in which the morphology is not considered would be, in our opinion, incomplete. Our data confirm that there are not morphological differences for the Spanish isolates. We have made some comments on the morphological results in Discussion section (lines (280-286).
Comment 6: This map needs improvement. There's no north, there's no scale and the quality is poor. I also recommend the addition of a smaller map showing where Spain is located within Europe.
Response 6: We have modified the figure as requested.
Comment 7: I suggest the authors add the numbers or the names of which sample or reagent was added within each of the wells in the figure. It'll improve the reading experience.
Response 7: The image has been changed according to other reviewer comment. We have modified the figure to add the sample identification to the gel lanes for an easier reading.
Comment 8: is there a prevalence on lagomorphs individuals? Since they are also within these wildcat's diet.
Response 8: We have not added any info about lagomorphs because all findings of the Hydatigera metacestode stage is reported in the scientific literature to occur in rodents, not in lagomorphs. We have added the following text in the discussion (lines 323-324): “ to the best of our knowledge, there are no records of Hydatigera metacestodes in lagomorphs.”
Finally, we have also corrected the typographical errors that the reviewer has indicated in the text (i.e., lines 182-236 Hydatigera Kamiyai, Table 4 (previously, table 5), 227-237).
Reviewer 3 Report
Comments and Suggestions for Authors
The paper titled "Identification of Hydatigera species in wildcats (Felis silvestris) from central Spain" identify Hydatigera species of this complex infecting wildcats in central Spain using both morphological and molecular methods. It is interesting to monitor wildcats. But I suggest the authors should sequnce 1-3 samples of Hydatigera species from 26 wildcats using cox1 gene to discuss the genetic diversity of Hydatigera. The use of multiplex primers for amplification to determine tapeworm infection is not recommended. Instead, it is advisable to directly sequence PCR products generated by amplification with species-specific primers. This approach enables precise identification of the tapeworm species infecting the 26 wildcats and facilitates comprehensive analysis of genetic variation across all parasite samples. Furthermore, it allows for the construction of a robust phylogenetic tree, thereby providing a more accurate and reliable basis for species delineation and taxonomic classification. Please ensure that the first letter of the family name is capitalized and that italics are not used; for abbreviated genus names, italicization should be applied.
L93-94,Check it. The magnification of the eyepiece must also be taken into account.
L96-106,L109-115 The two paragraphs have been integrated into a single cohesive paragraph to enhance textual continuity, improve logical flow, and meet formal requirements for conciseness and clarity in academic writing.
L122 add the complete PCR reaction system, eg dNTPs, Taq DNA polymerase...
L126 Were all PCR amplifications for the cox1, cytb, and nd4 genes were performed using an annealing temperature of 52 °C for 1 min?
L150 Table 1 Why were primers designed based on sequences retrieved from the NCBI database rather than being developed from newly generated sequencing data obtained in this study?
L173-174 How did the authors identify the Joyeuxiella species? No PCR primers targeting this parasite were reported in the manuscript, and there is no description of amplification or sequencing procedures used for its molecular confirmation.
L199 Figure 2 Figure 2 lacks clarity in terms of band size resolution, which limits the accurate interpretation of amplicon lengths. It is recommended that the authors sequence all PCR products to confirm the specificity of amplification. Alternatively, if the identification method is considered valid, a clear description of the validation approach should be provided to support its reliability.
L231 Table 4 could be moved to the supplementary materials.
L235 Table 5 could be moved to the supplementary materials.
For other minor revisions, please refer to the annotated PDF.

Author Response
Responses to reviewer 3
Comment 1: The paper titled "Identification of Hydatigera species in wildcats (Felis silvestris) from central Spain" identify Hydatigera species of this complex infecting wildcats in central Spain using both morphological and molecular methods. It is interesting to monitor wildcats. But I suggest the authors should sequence 1-3 samples of Hydatigera species from 26 wildcats using cox1 gene to discuss the genetic diversity of Hydatigera. The use of multiplex primers for amplification to determine tapeworm infection is not recommended. Instead, it is advisable to directly sequence PCR products generated by amplification with species-specific primers. This approach enables precise identification of the tapeworm species infecting the 26 wildcats and facilitates comprehensive analysis of genetic variation across all parasite samples. Furthermore, it allows for the construction of a robust phylogenetic tree, thereby providing a more accurate and reliable basis for species delineation and taxonomic classification. Please ensure that the first letter of the family name is capitalized and that italics are not used; for abbreviated genus names, italicization should be applied.
Thanks very much for the insightful comments and suggestions. As we estate in the material and methods section, we have done an initial genetic analysis (using the cox1) of 15 cestodes randomly chosen: we did the PCR amplification and sequencing to identify the species, and the sequences obtained (including one corresponding to H. parva, used as negative control for the multiplex PCR) have been uploaded to the GenBank database (sequences PV9763980-84). We indicate in the manuscript that the sequences of the 6 individuals corresponding to Hydatigera sp. were identical, while there were 3 haplotypes among the 9 individuals corresponding to H. kamiyai. Please note that the study focuses on the epidemiology of these cestodes in wildcats in Spain, not in the genetic heterogeneity of the parasite; for the purpose of this study, the developed multiplex PCR is valid as it allows identifying the species with a good time-cost relation and excellent reliability. As the reviewer indicates, if the study were to know the genetic diversity within each Hydatigera species (useful, for example, for species delineation and partition analysis for species discovery; or for relationships between specific haplotypes and pathology or sources of infection), each individual found should have been sequenced.
We have revised the use of italics in the manuscript for the genus-species names, and use the normal font for the family names.
Comment 2. L93-94,Check it. The magnification of the eyepiece must also be taken into account.
Response 2. The reviewer is right, we indicated the objective magnification, not the total magnification. We have corrected the values in the text (line 132); “… at 6.6-40x magnification”.
Comment 3. L96-106,L109-115 The two paragraphs have been integrated into a single cohesive paragraph to enhance textual continuity, improve logical flow, and meet formal requirements for conciseness and clarity in academic writing.
Response 3. We have modified the text accordingly.
Comment 4. L122 add the complete PCR reaction system, eg dNTPs, Taq DNA polymerase...
Response 4. We have not done the PCR mix; instead, we have used a commercial kit (PureTaq Ready-To-Go PCR Beads kit) which consists of a lyophilized bead containing all the components of the PCR reaction. We indicate this in the text (lines 160-161).
Comment 5. L126 Were all PCR amplifications for the cox1, cytb, and nd4 genes were performed using an annealing temperature of 52 °C for 1 min?
Yes. We indicated in the former version, and it is stated in L164 of the revised manuscript, the annealing temp for the initial cox1 PCR of 15 individuals, and once the species of these individuals were determined after sequencing, we set up a multiplex PCR using the same annealing temp (L176-188) and we indicate the conditions for the multiplex PCR. Please note that primers were sequence-specific, and template DNA was in each case from a single organism. The target sequences were enough different between species to ensure no cross-binding could be produced. This was checked against the 15 individuals used for the set-up of the method.
Comment 6. L150 Table 1 Why were primers designed based on sequences retrieved from the NCBI database rather than being developed from newly generated sequencing data obtained in this study?
Response 6. We need previously known sequences which we can use to design new primers. We cannot design primers specific for a given organism, if we do not know its DNA sequence. For this reason, we used the sequences available in GenBank; once aligned, we could search for sequence fragments having the adequate requirements (20 nt minimum length, with differences in the sequence in at least the terminal 3’ side of the primer between the sequence of the target species and other species’ sequences).
Comment 7. L173-174 How did the authors identify the Joyeuxiella species? No PCR primers targeting this parasite were reported in the manuscript, and there is no description of amplification or sequencing procedures used for its molecular confirmation.
The identification of the genus Joyeuxiella was carried out based on morphological characteristics using the taxonomic keys proposed by Khalil (1994). The differentiation from Hydatigera is easy (protrusible armed rostellum, double genital pore in Joyeuxiella, non-protrusible rostellum and single genital pore in Hydatigera). As this genus was not the objective of the present work, these individuals were not further processed.
Comment 8. L199 2 Figure 2 lacks clarity in terms of band size resolution, which limits the accurate interpretation of amplicon lengths. It is recommended that the authors sequence all PCR products to confirm the specificity of amplification. Alternatively, if the identification method is considered valid, a clear description of the validation approach should be provided to support its reliability.
Response 8. Figure 3 (Figure 2 in the former version of the manuscript) has been replaced as suggested by other reviewers. Please note that the set-up of the system included the specific identification of each of 15 individuals randomly chosen by using the (partial) cox1 gene. Once identified each species, one primer (forward primer HD) was designed to bind to the corresponding target sequence of all species, while two reverse primers (KHAR y HSR) were desgined to bind only the target sequence of H. kamiyai or Hydatigera sp., respectively. The JB3-JB4.5 (names have been changed from COD-COR, following another reviewer’s comment) were used in the same reaction, to check the amplification occur. If no specific bands were present, and the JB3-JB4.5 control band was neither present, then a “main” problem occurred that prevented the amplification reaction were obtained. If no specific bands were present but the JB3-JB4.5 control band was present, then the lack of specific bands were not related to problems in the amplification reaction but to template DNA corresponding to a different species (and, in such case, the amplicon would be sequenced to identify the species). For this purpose, the DNA of H. parva was used: in this case, the JB3-JB4.5 control band should be present, but the specific bands (corresponding to H. kamiyai or to Hydatigera sp.) should not. We think this is explained in the text (L193-198); to clarify the use of the JB3-JB4.5 primers, we have modified the text (lines 186-187) as follows: “The previously mentioned primers JB3 and JB4.5 were included in the reaction mix and used to amplify the cox1 fragment, serving as control of the reaction. “
Comment 9. L231 Table 4 could be moved to the supplementary materials.
Response 9. For comparison purposes, and considering the sensu lato species, we added this table to be able to compare our results (intensity of infection) with those already published. We have included published articles from around the world in which data on intensity of infection were indicated. After revising the manuscript, we consider it would be best to include this table in the body of the manuscript instead of moving it to supplementary materials. In the new version of the manuscript, it is Table 6.
Comment 10. L235 Table 5 could be moved to the supplementary materials.
Response 10: We think this table is enough informative and pertinent to keep it in the main text of the article. In the new version of the manuscript, it is now Table 1.
Comment 11. For other minor revisions, please refer to the annotated PDF.
Response 11. We have checked the comments in the pdf file and have made the corrections suggested.
Reviewer 4 Report
Comments and Suggestions for Authors
References have to be cited in the text using squared brackets style
L53 By authors used taxonomical terms are questionable. Better would be (mice and voles).
L70 authors should expand the description of cestode investigations in wildcats in Spain. I suggest a separate short paragraph on this topic. Now it is clear that cestodes were found, but not of Hydatigera genus; however, the prevalence, parasite species distribution, parasite load and applied methods for the discrimination of cestodes would be of particular interest. Also, author should provide in which parts of Spain previous studies have been conducted.
2.1. -2.5. titles have to be written using every first capitalised letter style. E.g., 2.2. Initial Processing and Cestode Recovery (words such as and, from, of etc. are not capitalised)
2.2.-2.4. please check formatting of the text, each new paragraph should be indented from the margin.
L107 please write exact magnification not objectives. microscope magnification is obtained by multiplying the eyepiece and objective magnifications
L114-115 for accurate PCRs the concentration of extracted DNA as well as purity of it is essential. If concentration is too high the dilutions have to be made. So, what was the range of DNA concentrations, is the DNA diluted? If so, to what concentration? How was concentration and purity of the DNA measured?
L141 it should be cytb
Table 1. it should be indicated that complete mtDNA sequences were compared.
L137-149 and Table 2. I could not understand the information here until I have looked at GenBank records and understanding that after cytb, NADH regions follow. Please revise the text that it would be clear in what principal primers for species diagnosis were developed.
L147-149 graphical view is necessary here to see that primers developed are suitable for the discrimination of the taxa selected. My suggestion is to provide alignment view and to demonstrate locations of primers binding sites
L151-153 why only Hydatigera parva was included as negative control, but in the Table 1 H. krepkogorski is also shown. What is the possibility to catch these parasite species in wildcats from Spain?
L117-156 authors have to provide concentration of DNA as well as primers (2 μl of primers has no meaning). At the moment, the repetition of PCRs in other laboratories based on description is complicated.
L160-161 authors should explain why. So, to provide approximate length of HD-COR fragment.
My suggestion for Table 2 is to provide the exact locations of each primer based on the selected reference sequence, which would make the description of the proposed method clearer.
L173-174 why sequences of Joyeuxiella sp. were not submitted to GenBank? What was the length of these sequences and authors have to provide BLAST analysis results in detail, the highest sequence similarity values.
L189-191 it is not clear; this sample was applied as negative control? Please explain in the text
L191-193 not all sequences, but only different haplotypes. This should be reflected in the text
Quality of the Figure 2 is not good, please improve it, now the figure is unacceptable for publication. My suggestion: enhance the photo so that the bands appear black and the entire background appears white. Also, authors should indicate lane numbers in the figure, also bands length for the DNA ladder used, and the exact ladder applied should be indicated.
L196 it should be 46.7%
L198 change to (Table 3 and Figure 1)
Table 4. Please correct the order of mean intensity and the range of the intensity
Table 4. So, there were no examinations of Hydatigera taeniaeformis s.l. in Lynx lynx?
L211-213 in which countries all three species have been identified, please list these countries, also if there are records of all three species in one host in certain country?
L220-221 please be specific, what exactly other markers?
L222 full gene names are not needed here, since authors above already have been using the abbreviated forms
L237 italic needed for the genus name
L260-261 figures do not correspond to those presented in the Table 3
for me not entirely clear from the text which studies presented in Tables 4 and 5 relied on molecular investigations and which solely on morphological examination? It would be good if this information were included in tables (e.g., it can be highlighted using asterixis)
Table 5 “Cricetidae, Muridae and Serbia” has to be written in normal style, please correct the formatting mistake
Author Response
Responses to reviewer 4
Comment 1: References have to be cited in the text using squared brackets style.
Response 1. We have revised the text and made the corrections where necessary.
Comment 2: L53 By authors used taxonomical terms are questionable. Better would be (mice and voles).
Response 2. We have changed “murines and microtines” by “mice and voles”, as suggested (lines 53).
Comment 3: L70 authors should expand the description of cestode investigations in wildcats in Spain. I suggest a separate short paragraph on this topic. Now it is clear that cestodes were found, but not of Hydatigera genus; however, the prevalence, parasite species distribution, parasite load and applied methods for the discrimination of cestodes would be of particular interest. Also, author should provide in which parts of Spain previous studies have been conducted.
Response 3. There are practically no studies on parasites in wildcats in Spain. The only references we have found, to the best of our knowledge, are a publication by Torres et al. (1989) (ref. 26 in the manuscript) and a conference paper by Gómez-Galindo et al. (2019) (ref. 27 in the manuscript). We have modified the text (lines 81-88) as follows: “There are almost no previous data on cestode infections in wildcats in Spain; the only data available (referred to H. taeniaeformis s.l.) are those obtained more than 30 years ago by Torres et al. (1989) in samples from all over Spain, who found a overall 60.3% prevalence; and those obtained between 2008-2015 by Gómez-Galindo et al. (2019) in samples collected in the south-east of Spain, who found a 36.8% prevalence. Other cestodes found in these studies included Taenia pisiformis, Joyeuxiella pasqualei, Diplopylidium nolleri and Mesocestoides spp. In both studies, identifications were based on morphological characters.”
Comment 4: 2.1. -2.5. titles have to be written using every first capitalised letter style. E.g., 2.2. Initial Processing and Cestode Recovery (words such as and, from, of etc. are not capitalised)
Response 4. We have revised the section titles and corrected them as indicated.
Comment 5: 2.2.-2.4. please check formatting of the text, each new paragraph should be indented from the margin.
Response 5. We have revised the manuscript and corrected the indentation when necessary.
Comment 6: L107 please write exact magnification not objectives. microscope magnification is obtained by multiplying the eyepiece and objective magnifications
Response 6. Thanks for the comment. We have revised the text and changed the values to magnification (“6.6-40x magnification”).
Comment 7: L114-115 for accurate PCRs the concentration of extracted DNA as well as purity of it is essential. If concentration is too high the dilutions have to be made. So, what was the range of DNA concentrations, is the DNA diluted? If so, to what concentration? How was concentration and purity of the DNA measured?
Response 7. We have not measured the DNA purity or its concentration, and we have used the total DNA undiluted. The points indicated by the reviewer are important if sequencing is to be done, specially if several organism sources of DNA can be present in the sample (i.e., if the DNA is extracted from a soil, water or faecal sample). In the present case, DNA was extracted from washed specimens, this eliminating contaminant sources of DNA. In case no PCR amplification were obtained from a sample, concentration and DNA:protein ration would have been determined and the PCR conditions (i.e., sample dilution) or the DNA extraction process, if necessary, would have been repeated. This was not the case for any of the cestodes we have analysed, and in all cases, PCR amplification were obtained.
Comment 8: L141 it should be cytb
Response 8: We have changed the text as indicated. citB-> cytb
Comment 9: Table 1. it should be indicated that complete mtDNA sequences were compared.
Response 9: Table caption (now Table 3) has been changed to “Complete mitochondrial sequences retrieved from GenBank utilized for multiplex PCR primer design.”
Comment 10: L137-149 and Table 2. I could not understand the information here until I have looked at GenBank records and understanding that after cytb, NADH regions follow. Please revise the text that it would be clear in what principal primers for species diagnosis were developed.
Response 10: The text has been changed as follows: “Each reaction simultaneously amplified two distant regions of mitochondrial DNA: the cox1 gene, used as internal control, and a diagnostic fragment including part of the cytochrome b gene (cytb) (~618 bp, for Hydatigera sp.), or encompassing the consecutive cytb-NADH dehydrogenase subunit 4 (nad4) genes (~1063 bp., for H. kamiyai). Based on complete mitochondrial DNA sequences of Hydatigera spp. …”
Comment 11: L147-149 graphical view is necessary here to see that primers developed are suitable for the discrimination of the taxa selected. My suggestion is to provide alignment view and to demonstrate locations of primers binding sites
Response 11: We have added a supplementary file containing the alignment of the mitochondrial sequences showing the location of the primers. This is indicated in Table 3 caption.
Comment 12: L151-153 why only Hydatigera parva was included as negative control, but in the Table 1 H. krepkogorski is also shown. What is the possibility to catch these parasite species in wildcats from Spain?
Response 12: As we indicated in response 3, there are almost no previous data from Spain, and H. krepkogorski has not been recorded in Spain. We have tried to obtain H. krepkogorski DNA, and we asked prof. Mitra Sharbatkhori, who recently (2018) analysed individuals of this species (doi: 10.30466/vrf.2018.33105), but he indicated that currently do not have available material from this species nor from H. taeniaeformis s.s. We also checked the possibility of obtaining H. taeniaeformis s.s. by contacting prof. Antti Lavikainen, but in this case, we had no response.
Comment 13: L117-156 authors have to provide concentration of DNA as well as primers (2 μl of primers has no meaning). At the moment, the repetition of PCRs in other laboratories based on description is complicated.
Response 13: We have added the primer concentration (lines 159-161): “Reactions were made in 25 µL containing 5 µL of template DNA and 2 µL of 5 pmol/µL solutions of each primer”. Also, in lines 183-186, “…two species-specific reverse primers were designed: one for H. kamiyai (HKAR; 5’-AARTAAAAACGTACCCAACTAGACAG) and one for Hydatigera sp. (HSR; 5’-ATTAATCTTATCATAACGACAACTAATAATCC) (all primers’ solutions at 5 pmol/µL)”. As mentioned in a previous response, we have not determined the DNA concentration prior to PCR amplification.
Comment 14:L160-161 authors should explain why. So, to provide approximate length of HD-COR fragment.
Response 14: We have modified the text as follows (lines 199-205): “Under these conditions, HD-JB4.5 amplification ( ~6660 positions) is not feasible because of insufficient time to complete the amplification.”
Comment 15: My suggestion for Table 2 is to provide the exact locations of each primer based on the selected reference sequence, which would make the description of the proposed method clearer.
Response 15: We have added a supplementary file 1 with the complete alignment of the mitochondrial sequences, in which we indicate the positions of the primers.
Comment 16: L173-174 why sequences of Joyeuxiella sp. were not submitted to GenBank? What was the length of these sequences and authors have to provide BLAST analysis results in detail, the highest sequence similarity values.
Response 16: We have not included this sequence because Joyeuxiella was not within the scope of the paper. Our intention was to focus on Hydatigera spp. and then Joyeuxiella was not further mentioned after indicating it was also present in three cases. Anyway, we have sequenced it (439 positions in the cox1 gene, primers JB3-JB4.5), obtaining a sequence which is about 93-94% similar (depending on the comparing sequence) to Joyeuxiella pasqualei. This data will be used in future publications.
Comment 17: L189-191 it is not clear; this sample was applied as negative control? Please explain in the text
Response 17: Hydatigera parva was used as control for the validity of the multiplex PCR, as we stated in the text (material and methods section, lines 190-192).
Comment 18: L191-193 not all sequences, but only different haplotypes. This should be reflected in the text
Response 18: We have modified the text (lines 251-253) as follows: “The sequences of H. parva and the haplotypes of H. kamiyai and ….”
Comment 19: Quality of the Figure 2 is not good, please improve it, now the figure is unacceptable for publication. My suggestion: enhance the photo so that the bands appear black and the entire background appears white. Also, authors should indicate lane numbers in the figure, also bands length for the DNA ladder used, and the exact ladder applied should be indicated.
Response 19: Figure 3 (Figure 2 in the former version of the manuscript) has been changed and labels have been added where necessary, following the recommendations.
Comment 20: L196 it should be 46.7%
Thank you for pointing out this mismatch. The value has been corrected to 46.7%.
Comment 21: L198 change to (Table 3 and Figure 1)
Response 21: The change has been made as suggested (Table 4 and Figure 1).
Comment 22: Table 4. Please correct the order of mean intensity and the range of the intensity
Response 22: The order has been changed as suggested in Table 6 (former Table 4).
Comment 23: Table 4. So, there were no examinations of Hydatigera taeniaeformis s.l. in Lynx lynx?
Response 23: We have added a reference in Table 6 (previously, Table 4) in relation to H. taeniaeformis in Eurasian lynx.
Comment 24: L211-213 in which countries all three species have been identified, please list these countries, also if there are records of all three species in one host in certain country?
Response 24: We have modified the text to indicate that the three species have been recorced in Europe (lines 65-80), and by considering the work by Lavikainen et al. (2016), who identified H. taeniaeformis s.s. in rodents in Spain, and our present results, the three species (H. taeniaeformis s.s., H. kamiyai and the European Hydatigera sp.) are present in our country (lines 352-356> CONCLUSIONS).
Comment 25: L220-221 please be specific, what exactly other markers?
Response 25: We have modified the text as follows: “… other molecular markers (mitochondrial 12S rRNA, NADH, nuclear 28S rRNA) [Okamoto 95, Lavikainen 16, Camacho-Giles 24, Miljevic 2023]”
Comment 26: L222 full gene names are not needed here, since authors above already have been using the abbreviated forms
Response 26: We have changed the full names by their abbreviations (cytb, nad4) where appropriate.
Comment 27: L237 italic needed for the genus name
Response 27: Changed.
Comment 28: L260-261 figures do not correspond to those presented in the Table 3 for me not entirely clear from the text which studies presented in Tables 4 and 5 relied on molecular investigations and which solely on morphological examination? It would be good if this information were included in tables (e.g., it can be highlighted using asterixis)
Response 28: In Table 6 (previously, Table 4) we presented the results from morphological studies (thus, H. taeniaeformis s.l.). We have modified the paragraph now mentioned by the reviewer as follows: “The parasite burden of Hydatigera spp. observed in wildcats in the present study ranged from 4 to 36 cestodes per individual host, with H. kamiyai showing the widest range (1 to 16 cestodes per host). There are no reports on intensity of infection based on genetic data; Mederle et al. [NUMERO] indicate the presence of H. kamiyai (identified by them as T. taeniaeformis, sequence EU219554) in 3 out of 7 wildcats, but they did not indicate the number of individuals per animal. Our results, combining H. kamiyai and Hydatigera sp., are consistent with the data published in other studies based only in morphological analyses (thus considering H. taeniaeformis s.l.), in which intensity of infection varied in the range 2-20 in wildcats in Germany [ref. 28 de table 4 vieja] and between 1-79 in domestic cats in other parts of the world [refs. 31, 36, 41, 42 de la table 4 vieja]. “
Comment 29: Table 5 “Cricetidae, Muridae and Serbia” has to be written in normal style, please correct the formatting mistake
Response 29: Corrected as suggested.
Round 2
Reviewer 1 Report
Comments and Suggestions for Authors
Dear authors,
I would like to commend the introduction and thank you for taking my comments into consideration. This has greatly strengthened your manuscript – the introduction is now clear, and readers have a more detailed overview of what is happening among the species of the genus Hydatigera, as recent taxonomic changes have indeed made understanding more complex. I am also glad that you highlighted the importance of your study area for this new species and related aspects. By additionally emphasizing the age of previous data in the introduction, you have further highlighted the significance of your own data.
Follow-up to Response 2:
With this explanation, you have considerably improved the manuscript, especially since you have provided a new and well-prepared gel image with clearly visible bands, where the control band can now be clearly seen — unlike in the previous image, which was not sufficiently informative and made it difficult for me to understand certain parts of your study (the control band was not visible).
Also, you are right — the correction and addition of the horizontal line in Table 2 are important, as their previous absence caused confusion. Everything is now clear, and this line greatly improves the clarity, allowing one to easily identify which species corresponds to each GenBank accession number.
Follow-up to Response 2:
All right, it is important to present it that way and to emphasize the possibility of sequencing the control band.
Follow-up to Response 3:
Alright, I appreciate that.
Follow-up to Response 4:
I appreciate the clarification.
Follow-up to Response 5
The correction has been properly addressed.
Follow-up to Response 6:
I appreciate this correction, as it is one of the most important for understanding the study.
Follow-up to Response 7:
I greatly appreciate that you performed the experiment once again and mixed two species to demonstrate that co-infection (H. kamiyai and Hydatigera sp.) can be detected in a single sample. However, as you state: "As you can see in the next image, the bands corresponding to each species are present in the mixed DNA. The first lane contains the molecular marker, lanes 2 and 3 (mix1, mix2) correspond to amplification of DNA from mixed proglottids, and lanes 4 and 5 correspond to amplification of DNA from H. kamiyai (Hk) and European Hydatigera sp. (Hsp)." — I must apologize, but I do not see this clearly anywhere. Figure 3 has indeed been improved, and I have already commended that, but essentially it remains very similar to the first version.
In any case, it is useful to know that it is possible to see all three bands (H. kamiyai, Hydatigera sp., and the control band) in a single lane, even though I understand your concern that some species could thus be "missed/unidentified." This information is particularly important for researchers who may use this method solely to distinguish the H. taeniaeformis sensu lato complex, in cases where neither H. parva nor H. krepkogorski are expected.
Follow-up to Response 8:
I appreciate this correction.
Follow-up to Response 9:
I appreciate this correction.
Follow-up to Response 10:
I appreciate this correction.
Follow-up to Response 11:
The correction has been properly addressed.
Follow-up to Response 12:
I’m glad you recognized this, as it was a serious mistake and it changes the entire flow of your discussion.
Follow-up to Response 13:
Ok, I still believe that this is not necessary. If you wish to keep the table, the title could indicate that it refers to different cat species. Since you want to emphasize this information, the title should be consistent with that.
Example: Table 6. Epidemiological data of Taenia taeniaeformis/Hydatigera taeniaeformis s.l. across Felidae species
Follow-up to Response 15:
I appreciate this correction. You have now better highlighted the significance of your research and the importance of your European Hydatigera sp., as well as, more generally, the relevance of your study area for this species that occurs here.
Additionally:
Row 194–195: Please add the molarity of the primers.
Row 206: Table 3: Should cytb be listed under JB3/JB4.5 in “Primer location in mtDNA,” or should it remain only with cox1?
Row 229: I commend the images of the parasites and the table with the analysis and morphological measurements. If possible, please indicate the number of individuals analyzed to obtain the reported values.
Row 273-279: This paragraph: 'We were not able to analyse the internal structures of the mature proglottids (e.g., testes, ovaries, cirrus sac) as they appeared partially degenerated. This deterioration is likely related to the post-mortem condition of the hosts, which were found dead and subsequently frozen until necropsy could be performed. Freezing and prolonged post-mortem intervals may have affected the preservation of these structures, whereas more robust features such as the scolex (rostellum, suckers, hooks) and uterine branches remained in a better condition and could be measured accurately.' — please condense this into a single sentence, as it is currently too detailed and unnecessary. Also, link it in some way to the following discussion: 'The three species comprising the H. taeniaeformis complex exhibit very similar morphology...
Row 298: Only comment: How nicely you have now explained the role of the control band and how you addressed certain "limitations" – I truly commend this.
Regards,
Author Response
Responses to Reviewer 1
We would like to thank the reviewer for the cordial tone and the scientific quality of their comments. In this second revision, there are only two observations to address, in the follow-up to responses 7 and 13, as well as some additional comments.
Follow-up to response 7
I greatly appreciate that you performed the experiment once again and mixed two species to demonstrate that co-infection (H. kamiyai and Hydatigera sp.) can be detected in a single sample. However, as you state: "As you can see in the next image, the bands corresponding to each species are present in the mixed DNA. The first lane contains the molecular marker, lanes 2 and 3 (mix1, mix2) correspond to amplification of DNA from mixed proglottids, and lanes 4 and 5 correspond to amplification of DNA from H. kamiyai (Hk) and European Hydatigera sp. (Hsp)." — I must apologize, but I do not see this clearly anywhere. Figure 3 has indeed been improved, and I have already commended that, but essentially it remains very similar to the first version.
In any case, it is useful to know that it is possible to see all three bands (H. kamiyai, Hydatigera sp., and the control band) in a single lane, even though I understand your concern that some species could thus be "missed/unidentified." This information is particularly important for researchers who may use this method solely to distinguish the H. taeniaeformis sensu lato complex, in cases where neither H. parva nor H. krepkogorski are expected.
Response:
The figure we referred to ("you can see in the next image") is not a figure from the manuscript, but rather a photograph of a gel showing the banding patterns obtained from the PCR analysis of proglottids from mixed species. This image was prepared specifically for your reference and was not intended for inclusion in the manuscript.
We drafted our responses in a Word document and then copied and pasted the text into the journal's web form. We were not aware that the image failed to copy during this process. We have now submitted the responses as a PDF file to ensure the image is kept (see immediately below).

We have not included this image in the manuscript because the scenario it depicts is not representative of our study: proglottids from individual cestodes were analyzed, meaning that mixed bands (i.e., mixed species) would not be present in the gel lanes. Furthermore, as samples from H. taeniaeformis s.s. were not analyzed, we cannot propose this method as a tool for differentiating species within the H. taeniaeformis complex, nor can we suggest its use for processing pooled proglottids. Under the conditions of our study, these were not valid objectives.
Follow-up to Response 13:
Ok, I still believe that this is not necessary. If you wish to keep the table, the title could indicate that it refers to different cat species. Since you want to emphasize this information, the title should be consistent with that.
Example: Table 6. Epidemiological data of Taenia taeniaeformis/Hydatigera taeniaeformis s.l. across Felidae species
Response: We have modified the table title as suggested.
Row 194–195: Please add the molarity of the primers.
Response: We have added the molarity (5 µM)
Row 206: Table 3: Should cytb be listed under JB3/JB4.5 in “Primer location in mtDNA,” or should it remain only with cox1?
Response: The reviewer is right, this is a mistake; JB3/JB4.5 are located on cox1, not in cytb. The table has been corrected.
Row 229: I commend the images of the parasites and the table with the analysis and morphological measurements. If possible, please indicate the number of individuals analyzed to obtain the reported values.
Response: As detailed in the Materials and Methods section, all specimens with a scolex were stained to identify cestodes belonging to the Hydatigera taeniaeformis complex. To assess potential morphological differences between the species within this complex (H. kamiyai and Hydatigera taeniaeformis), morphological characteristics (number, size, and arrangement of rostellar hooks; morphology of mature and gravid segments) were examined in 20 individuals of each species, after confirmation by molecular analysis. This information has been incorporated into section 2.3 of the Materials and Methods, line 141.
Row 273-279: This paragraph: 'We were not able to analyse the internal structures of the mature proglottids (e.g., testes, ovaries, cirrus sac) as they appeared partially degenerated. This deterioration is likely related to the post-mortem condition of the hosts, which were found dead and subsequently frozen until necropsy could be performed. Freezing and prolonged post-mortem intervals may have affected the preservation of these structures, whereas more robust features such as the scolex (rostellum, suckers, hooks) and uterine branches remained in a better condition and could be measured accurately.' — please condense this into a single sentence, as it is currently too detailed and unnecessary. Also, link it in some way to the following discussion: 'The three species comprising the H. taeniaeformis complex exhibit very similar morphology...
Response: we have modified the paragraph and added it to the next one, as follows: "Some internal structures (e.g., testes, ovaries, cirrus sac) appeared partially degenerated, likely due to the hosts’ post-mortem condition and freezing prior to necropsy, although more robust features such as the scolex and uterine branches were well preserved and could be measured accurately. The morphological characteristics of these structures did not allow differencing the adult stages of H. kamiyai and the European Hydatigera sp., a result which is in accordance with previous studies that stated the three species comprising the H. taeniaeformis complex exhibit very similar morphology [21]".
Reviewer 2 Report
Comments and Suggestions for Authors
The authors have adequately addressed all of the previous suggestions. In my opinion, the manuscript is now suitable for publication. My only remaining recommendation is to include a scale bar in kilometers on both maps, so that readers can clearly understand the spatial scale represented.
Author Response
Review Report 2.
The authors have adequately addressed all of the previous suggestions. In my opinion, the manuscript is now suitable for publication. My only remaining recommendation is to include a scale bar in kilometers on both maps, so that readers can clearly understand the spatial scale represented.
Response: We sincerely thank the reviewer for the positive evaluation and the helpful suggestion. Following the recommendation, we have added a scale bar in kilometers to both maps so that readers can clearly understand the spatial scale represented. The revised figures have been updated accordingly in the manuscript.
Reviewer 3 Report
Comments and Suggestions for Authors
The manuscript has been improved compared to the initial version. However, minor revisions are required at lines 67, 69, and 84, where inconsistencies in the citation formatting were identified. Specifically, two cited references include inappropriate time formats that should be removed. Upon completion of these revisions, the manuscript will be suitable for acceptance.
Author Response
Review Report 3.
The manuscript has been improved compared to the initial version. However, minor revisions are required at lines 67, 69, and 84, where inconsistencies in the citation formatting were identified. Specifically, two cited references include inappropriate time formats that should be removed. Upon completion of these revisions, the manuscript will be suitable for acceptance.
Response: We sincerely thank the reviewer for carefully checking the manuscript and for pointing out the inconsistencies in the citation formatting (lines 67, 69, 83 and 84). These issues have been corrected in the revised version, and the inappropriate time formats have been removed to ensure consistency with the journal’s citation style.
Reviewer 4 Report
Comments and Suggestions for Authors
L71 please specify, here authors are referring to H. taeniaeformis species complex? Slight modification of this sentence is needed
Author Response
Review Report 4.
L71 please specify, here authors are referring to H. taeniaeformis species complex? Slight modification of this sentence is needed
Response: We thank the reviewer for this valuable comment. In this part of the text, we were referring to two putative Hydatigera species recently identified in China from rodent hosts. However, as indicated later in line 79, the adult cestodes of these Asian Hydatigera spp. have not yet been described. Therefore, since only the larval stages are known and no morphological data are available, it is not possible to determine whether they belong to the Hydatigera taeniaeformis species complex. To improve clarity, we have modified the text accordingly. Specifically, we have incorporated the information that was previously stated in line 79 into the paragraph describing the Hydatigera taeniaeformis complex. The revised text now reads as follows:
“However, these have only been described from larval stages, and the adult cestodes of the Asian Hydatigera spp. have not yet been described; therefore, their inclusion within the Hydatigera taeniaeformis complex remains uncertain.”